# Decreased Indian Ocean Dipole variability under prolonged greenhouse warming

**Soong-Ki Kim** [1], **Hyo-Jin Park**[1,2], **Soon-Il An** [1,2,3] ✉, **Chao Liu** [1],
**Wenju Cai** [4,5,6,7], **Agus Santoso**[8,9,10] & **Jong-Seong Kug** [11]

The Indian Ocean Dipole (IOD) is a major climate variability mode that substantially influences weather extremes and climate patterns worldwide. However, the response of IOD variability to anthropogenic global warming remains highly uncertain. The latest IPCC Sixth Assessment Report concluded that human influences on IOD variability are not robustly detected in observations and twenty-first century climate-model projections. Here, using millennial-length climate simulations, we disentangle forced response and internal variability in IOD change and show that greenhouse warming robustly suppresses IOD variability. On a century time scale, internal variability overwhelms the forced change in IOD, leading to a widespread response in IOD variability. This masking effect is mainly caused by a remote influence of the El Niño–Southern Oscillation. However, on a millennial time scale, nearly all climate models show a long-term weakening trend in IOD variability by greenhouse warming. Our results provide compelling evidence for a human influence on the IOD.

The Indian Ocean Dipole (IOD) is a leading climate variability mode in the tropical Indian Ocean[1–3]. A positive IOD phase features anomalously warm sea surface temperature (SST) in the western Indian Ocean and cold SST in the east, and the zonal warm-cold pattern is reversed during its negative phase. IOD events alter ocean and atmospheric circulation patterns[3–5] and the associated precipitation anomalies cause substantial impact on Indian Ocean-rim countries, such as droughts and floods in Australia[6,7], Southeast Asia[8–10], the Indian continent[11,12], and East Africa[13], wildfires in southeast Australia[14] and Indonesia[9], coral reef dieback in western Sumatra[15], and malaria outbreaks in East Africa[16].

Understanding the response of IOD to global warming has been one of the long-standing important problems in climate science[4,5,17–20]. The latest IPCC Sixth Assessment Report (AR6)[21] has concluded that the human influence on IOD is not robustly detected during the observational period[21,22] and the forced change in IOD by global warming in the future is uncertain due to a lack of robust evidence[21,23]. IOD variability, a fluctuation in climate variables induced by the IOD, is typically measured by the dipole mode index (DMI), which is defined as the difference between the western (50° E–70° E and 10° S–10° N) and eastern (90° E–110° E and 10° S–0° N) SST anomalies of the tropical Indian Ocean. This traditional IOD index serves as a fundamental

[1]Irreversible Climate Change Research Center, Yonsei University, Seoul, Republic of Korea. [2]Department of Atmospheric Sciences, Yonsei University, Seoul, Republic of Korea. [3]Division of Environmental Science and Engineering, Pohang University of Science and Technology (POSTECH), Pohang, Republic of Korea. [4]Frontiers Science Center for Deep Ocean Multispheres and Earth System/Physical Oceanography Laboratory/Sanya Oceanographic Institution, Ocean University of China, Qingdao, China. [5]Laoshan Laboratory, Qingdao, China. [6]State Key Laboratory of Marine Environmental Science & College of Ocean and Earth Sciences, Xiamen University, Xiamen, China. [7]State Key Laboratory of Loess and Quaternary Geology, Institute of Earth Environment, Chinese Academy of Sciences, Xi'an, China. [8]Centre for Southern Hemisphere Oceans Research (CSHOR), CSIRO, Hobart, Australia. [9]Climate Change Research Centre and Australian Research Council (ARC) Centre of Excellence for Climate Extremes, The University of New South Wales, Sydney, Australia. [10]International CLIVAR Project Office, Ocean University of China, Qingdao, China. [11]School of Earth and Environmental Sciences, Seoul National University, Seoul, Republic of Korea. ✉e-mail: sian@yonsei.ac.kr

physical metric for IOD dynamics and associated teleconnection impacts. So far, no significant trends in the DMI variability have been detected in either observations or future climate-model projections[17,18], despite a strong warming signal being present in the tropical Indian Ocean[18,24]. Although some studies founded changes in the intensity and frequency of strong positive IOD events in the projected future climate using the principal component decomposition technique on SST[5] and rainfall[4,25], the response of DMI variability, which describes the warm-cold cycles of the tropical Indian Ocean SST anomalies, remains uncertain and no inter-model consensus has yet been found.

The lack of robust changes in DMI variability appears contradictory to our understanding of IOD dynamics. The mean state of the tropical Indian Ocean, which is known to physically shape the IOD feedback processes and variability, shows robust changes due to global warming: faster warming in the west than east (i.e., positive IOD-like warming), shoaling of the eastern thermocline, and easterly wind anomalies[18]. Such a clear warming pattern in the tropical Indian Ocean raises the question as to why the change in IOD amplitude is not as robust as the change in the background mean state.

One of the possible reasons for this problem is internal variability, a natural variation of IOD characteristics caused by intrinsic atmospheric and oceanic processes. Internal variability can mask forced changes in IOD variability due to global warming, resulting in a wide range of IOD amplitude projections and an obscured link with the changes in the mean background state of the tropical Indian Ocean. A large ensemble simulation using the Community Earth System Model shows that internal variability alone can generate widespread long-term trends in future IOD variability change; projected IOD amplitude in the twenty-first century either increases or decreases depending on the ensemble member with different internal variability, despite the same warming forcing[26]. Notably, a spectral analysis of the CMIP6 preindustrial control (piControl) experiment shows a surprisingly low-frequency peak in the inter-centennial IOD amplitude change, ranging from 1/400 year$^{-1}$ to 1/50 year$^{-1}$ (Supplementary Fig. 1). This implies that the internal variability has the potential to influence future CMIP6 IOD projections, which are typically assessed over an 86-year time frame (2015-2100). Taken together, this underscores the need for an unprecedented long length of warm climate simulations to reliably resolve forced response and internal variability.

In this Article, we utilize climate simulations from the Long Run Model Intercomparison Project[27] (LongRunMIP), an archive of millennial-length scale fully coupled climate-model simulations, to clarify a forced response of IOD to greenhouse warming. The Long-RunMIP provides global climate simulations subject to a wide range of atmospheric $CO_2$ levels with a length of typically longer than 1000 years. The typical forcing scenario is an instantaneous doubling (abrupt2x), quadrupling (abrupt4x), and octupling (abrupt8x), and a gradual increase at a rate of 1% per year until the atmospheric $CO_2$ level doubles (1pct2x) and quadruples (1pct4x). After the $CO_2$ increase, the forcing is stabilized for typically longer than 1000 years. The Long-RunMIP also provides a preindustrial control simulation with constant external forcings (control). Therefore, the LongRunMIP provides a unique opportunity to disentangle forced response and internal variability and to systematically study the response of IOD to different levels of greenhouse warming (e.g., ref. 28).

We evaluate the IOD simulation performance of all available models from the LongRunMIP archive, and select 9 models that can reasonably simulate the observed temporal and spatial characteristics of IOD ("Methods" section and Supplementary Discussion 1). Our analysis sample includes 9 control and 18 high-$CO_2$ simulations (total 27) from 9 models (Supplementary Table. 1). We use the DMI, a commonly used index for IOD, to describe IOD variability ("Methods" section). We apply two different measures for the strength of the IOD variability. The main metric is the IOD amplitude, defined as the

standard deviation of the DMI. The other is IOD event intensity, defined as the peak DMI during the positive or negative IOD event ("Methods" section). Both metrics are highly correlated by definition (i.e., if the IOD amplitude is high, the IOD event intensity is very likely to be high, and vice versa), but we mainly use the IOD amplitude to examine the change in the IOD variability strength. The IOD amplitude is a fundamental metric that can show the overall response of the IOD variability to greenhouse gas warming. The simplicity of the IOD amplitude definition has advantages that facilitate the physical interpretation of the results.

## Results

### Equilibrium IOD response

We first analyze the equilibrium response of IOD variability to $CO_2$ forcing. We use the last 500 years of variables where the global mean surface temperature (GMST) reaches nearly equilibrium (Supplementary Fig. 2) and define this time frame as the equilibrium period. The simulation array of different levels of equilibrium GMST allows us to examine the forced equilibrium response of IOD to greenhouse warming. We calculate the IOD amplitude for the equilibrium period of each control and high-$CO_2$ simulation.

In almost all models, the IOD amplitude decreases monotonically with increasing $CO_2$ (Fig. 1a). The exceptions are HadCM3L and MIROC3.2. HadCM3L simulates a decreased IOD amplitude in the abrupt4x, but an increased IOD amplitude in the abrupt2x and abrupt8x simulations. MIROC3.2 shows a slight increase in 1pct2x, but a large decrease in 1pct4x. Simulation-wise, the majority of the high-$CO_2$ simulations (15 out of 18) show decreased IOD amplitude compared to their control simulations. The average intensity of positive and negative IOD events decreases in nearly all high-$CO_2$ simulations (18 of 18 high-$CO_2$ simulations for positive IOD and 15 of 18 for negative IOD), and the change is monotonic with $CO_2$ in most models (Supplementary Fig. 3 and "Methods" section). Both the reduced SST variability in the eastern and western tropical Indian Ocean contribute almost equally to the decrease in IOD amplitude (Supplementary Fig. 4). The seasonal variability of IOD decreases in almost all seasons (Supplementary Fig. 5). These changes show that the weakening of IOD variability is robust under equilibrium greenhouse warming. The equilibrium sensitivity analysis ("Methods" section) suggests that the change in IOD amplitude per 1 °C increase in GMST is estimated to range from −6.6% to 0.6%, and 8 out of 9 models show a negative sensitivity (Fig. 1b).

Notably, the CESM1.0.4 simulates a substantial weakening of IOD variability, that is a near-collapse response to $CO_2$ forcing. The power spectra of the IOD variability in a broad frequency range (1/50 to 1/2 year$^{-1}$) largely decreases with increasing $CO_2$ levels (Supplementary Fig. 6). In particular, the spectral power of the IOD variability is almost indistinguishable from the climate noise level in the abrupt8x simulation. This suggests that the IOD mode is in a near-collapse state and that the tropical Indian Ocean SST variability is dominated by climate noise.

Such transition occurs not only in the amplitude but also in the seasonal phase locking and spatial pattern of the CESM1.0.4 simulations. The IOD commonly peaks in September in all CESM1.0.4 simulations, but a secondary peak appears in April with increasing $CO_2$ levels (Supplementary Fig. 5), indicating an alteration of the typical seasonality of the IOD. This alteration in the seasonality also occurs in other models including HadCM3L and CCSM3II. HadCM3L simulations show a transition in seasonal peak from October (control and abrupt2x) to July (abrupt4x and abrupt8x). CCSM3II simulations show a gradual shift of seasonal peak from September (control) to August (abrupt700ppm) to June (abrupt1400ppm). Such a pronounced increase in IOD variability in boreal summer is consistent with the previous study[19] shows an increased occurrence of early positive IOD events by global warming

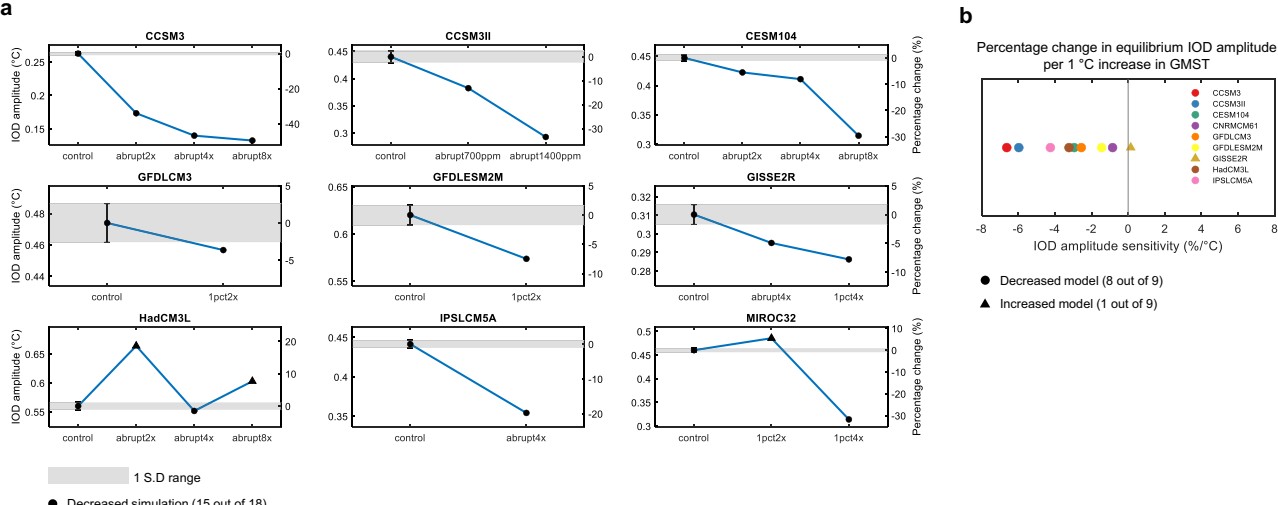

**Fig. 1 | Changes in Indian Ocean Dipole (IOD) amplitude in a warm equilibrium climate. a** Changes in IOD amplitude in the equilibrium period where global mean surface temperature (GMST) reaches equilibrium after an increase of atmospheric $CO_2$ level. The absolute IOD amplitude (left axis) and the percentage change from the control simulation (right axis). A one standard deviation range of IOD amplitude in the control simulation is shown in gray shading. A high-$CO_2$ simulation with a decrease (increase) in IOD amplitude outside of the one standard deviation range of the control simulation is marked with a black circle (triangle). In all high-$CO_2$ simulations, the IOD amplitude change compared to the control simulation is statistically significant ($P < 0.05$). The statistical significance is tested with the $F$-test for a pair of dipole mode index (DMI) time series of the control and high-$CO_2$ simulations. **b** The equilibrium sensitivity of IOD amplitude to GMST. For each model, the equilibrium sensitivity is calculated using an array of IOD amplitude and GMST in the equilibrium period ("Methods" section). The model with negative (positive) sensitivity is marked with a colored circle (triangle). This equilibrium sensitivity indicates the percentage change in IOD amplitude per 1 °C increase in GMST under an equilibrium climate condition.

under the IPCC's high-emissions scenario (although the warming scenario is different).

The spatial pattern of variability in CESM1.0.4 simulations also change from a dipole-like to a monopole-like pattern due to the weakening of the western Indian Ocean SST variability (Supplementary Fig. 7). The abrupt8x shows a clear monopole-like variability pattern, which is more appropriately referred to as Indian Ocean Monopole rather than an IOD. These results show that the main characteristics of the IOD in a significantly warm climate can substantially deviate from those of what are currently known about the IOD. This is somewhat analogous to the bifurcation or critical transition phenomenon[29,30], which refers to qualitative changes in the characteristics of a system when it crosses the critical threshold.

**Transient IOD response**

Next, we analyze the transient response of the IOD amplitude to the $CO_2$ forcing. We calculate the 100-year moving IOD amplitude for the entire simulation period. This window length corresponds to the typical time length for assessing future IOD change (e.g., comparing IOD variability for 1900–1999 and 2000–2099). The inter-simulation mean of the equilibrium GMST warming level of the quadrupling $CO_2$ simulations (abrupt4x and 1pct4x) is 5.5 °C (Supplementary Fig. 2). This is comparable to the projected warming level over 2081–2100 in the IPCC's high-$CO_2$ emissions scenario[21], which ranges from 3.3 °C to 5.7 °C, although the detailed warming pathway is different.

The evolution of IOD amplitude against GMST level is shown in Fig. 2a. In nearly all high-$CO_2$ simulations (17 out of 18), the IOD amplitude shows a long-term decreasing trend with increasing GMST. The exception is HadCM3L abrupt8x. This is consistent with the equilibrium response shown in Fig. 1, indicating that the long-term trend can be read as a forced response due to greenhouse warming towards the equilibrium state. However, within this long-term trend, the IOD amplitude largely fluctuates, not in line with the steadily increasing GMST. For example, the CESM1.0.4 abrupt4x simulation shows that the IOD amplitude increases in the GMST range between

18 °C and 19 °C, contrary to the long-term decreasing trend (Fig. 2a). This fluctuation is due to internal variability of the IOD amplitude, a deviation from the forced long-term trend. Throughout the paper, we refer to the anomalous short-term fluctuation in the variable that deviates from the long-term forced change as internal variability. This shows that the internal variability can mask the forced response to greenhouse warming by broadening the response range of the IOD amplitude.

To quantify the internal variability of IOD amplitude deviating from its forced long-term trend, we perform the transient sensitivity analysis ("Methods" section). Based on the results shown in Fig. 2a, we calculate the moving transient sensitivity of IOD amplitude against GMST and derive its distribution. This distribution represents an ensemble of all possible measurable transient sensitivities of IOD amplitude under a given warming forcing. If the change in IOD amplitude is driven solely by the forced response and not influenced by internal variability, the distribution would converge to the long-term trend sensitivity. In all high-$CO_2$ simulations, the transient sensitivity shows widespread distribution that deviates from its long-term trend sensitivity (Fig. 2b and Supplementary Fig. 8). The distribution range of the transient sensitivity differs between the model and simulations. For example, the CCSM3II abrupt700ppm ranges from −92%/°C to 50%/°C and the IPSL-CM5A-LR abrupt4x ranges from −37%/°C to 13%/°C (mean minus/plus one standard deviation). However, in all high-$CO_2$ simulations, the transient sensitivity ranges from negative to positive value, and its lower/upper bound is much larger than its long-term trend sensitivity.

Consistent with the IOD amplitude change, the intensities of both positive and negative IOD events also show long-term decreasing trends with increasing GMST (15 of 18 high-$CO_2$ simulations for positive IOD and 15 of 18 for negative IOD), but show large fluctuations and widespread transient sensitivity to GMST (Supplementary Figs. 9 and 10).

This suggests that internal variability significantly influences the inter-centennial IOD variability change to GMST increase and masks

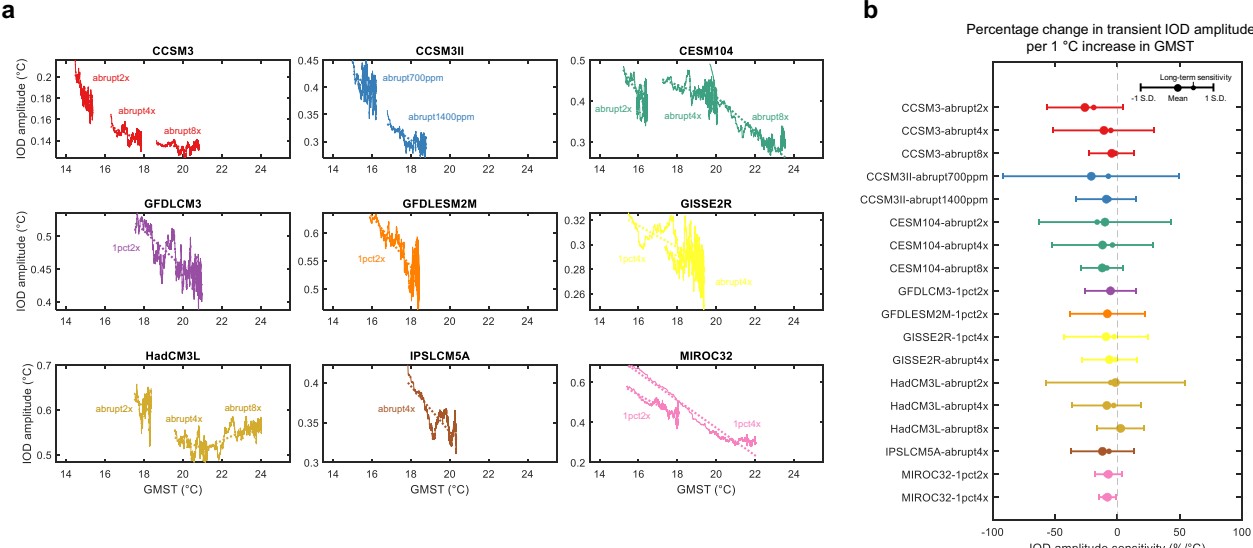

**Fig. 2 | Evolution of the Indian Ocean Dipole (IOD) amplitude under greenhouse gas warming. a** Evolution of the 100-year moving IOD amplitude against the global mean surface temperature (GMST) change. The linear regression line of IOD amplitude against GMST is shown as a dashed line. All high-$CO_2$ simulations show a decreasing trend in IOD amplitude with increasing GMST, except for HadCM3L abrupt8x. All linear trends are statistically significant ($P < 0.05$). **b** The statistics of the transient sensitivity of the IOD amplitude to GMST. For each high-$CO_2$ simulation, the transient sensitivity is calculated with moving GMST window of 1 °C ("Methods" section). The mean and one standard deviation of the transient sensitivity distribution is shown as a large black circle and error bar. The full distribution of the transient sensitivity can be found in Supplementary Fig. 8. The long-term sensitivity (i.e., the slope of the linear regression line in **a**) is plotted as a small black circle. The distribution represents an ensemble of all possible measurable transient sensitivities of the IOD amplitude under increasing GMST.

the long-term response. The masking effect of internal variability becomes stronger with increasing strength and decreasing frequency of internal variability (Supplementary Discussion. 2).

## Physical sources of the IOD change

To investigate the physical sources of the forced change and internal variability of IOD amplitude, we utilize a simple model for the IOD[20,31,32] ("Methods" section). The simple IOD model includes three key parameters/terms that characterize IOD variability, namely, local feedback processes ($\lambda$), external El Niño–Southern Oscillation (ENSO) forcing ($\beta \times ENSO$), and weather noise ($\sigma$). The simple IOD model, which reasonably reproduces the IOD variability (blue lines in Fig. 3f, g), disentangles the key physical sources of IOD as each parameter ("Methods" section and Supplementary Figs. 11–14) and allows us to quantify the contribution of each of physical source to the change in IOD amplitude. All three parameters/terms, $\lambda$, $\beta \times ENSO$, and $\sigma$ are robustly decreased in the equilibrium period of high-$CO_2$ simulations (Fig. 3a and Supplementary Fig. 11) and exhibit considerable internal variability (Fig. 3e and Supplementary Figs. 12–14) ("Methods" section).

We perform two different sets of experiments with the simple IOD model. First, we perform the parameter perturbation experiment ("Methods" section) to quantify how much each forced change in $\lambda$, $\beta \times ENSO$, and $\sigma$ contributes to the forced change in the IOD amplitude. Second, we perform the low-pass parameter experiment ("Methods" section) to quantify how much each of the internal variability of $\lambda$, $\beta \times ENSO$, and $\sigma$ contributes to the internal variability of the IOD amplitude.

The results of the parameter perturbation experiment (Fig. 3f) show that the decrease in $\lambda$ is the most dominant factor for the forced IOD amplitude change, and the decrease in $\sigma$ is a secondary contributor. The decrease in $\beta \times ENSO$ has a minimal contribution. This shows that greenhouse warming suppresses the local feedback processes in the tropical Indian Ocean, leading to the decrease in IOD amplitude. However, the weakened ENSO forcing makes virtually no contribution to the IOD change, even though ENSO is known to be a major forcing of IOD[31,33]. The change in local feedback processes is

associated with the changes in mean climatological state of the tropical Indian Ocean[17,18,20] (detailed discussion can be found in Supplementary Discussion 3).

In contrast, the results of the low-pass experiments (Fig. 3g) show that the internal variability in $\beta \times ENSO$ is the most dominant factor for the internal variability of the IOD amplitude, and $\lambda$ and $\sigma$ play secondary roles. This indicates that the inter-centennial change in ENSO forcing is a primary source of the internal variability of IOD amplitude, while the local feedback processes have a lesser influence. Thus, the masking effect of internal variability, which leads to the widespread response of IOD amplitude, is mainly attributed to the remote influence of ENSO.

The remote ENSO forcing is a combined effect of ENSO variability (*ENSO*) and sensitivity of IOD to ENSO variability ($\beta$). We perform additional experiments to separately quantify their contribution to the forced change and the internal variability of the IOD amplitude ("Methods" section). The changes in both *ENSO* and $\beta$ make virtually no contribution to the forced decrease in IOD amplitude (Supplementary Fig. 15). The decrease in *ENSO* variability makes a larger contribution than $\beta$, but the contribution is very small compared to $\lambda$ and $\sigma$. The internal variability of *ENSO* alone makes a considerable contribution to the internal variability of the IOD amplitude, similar in magnitude to that of $\lambda$ and $\sigma$, and $\beta$ makes a slightly smaller contribution (Supplementary Fig. 16). This shows that the large internal variability of $\beta \times ENSO$ and its influence on IOD is attributed to combined internal variability of *ENSO* and $\beta$.

In summary, these results from the simple IOD model experiments show that the internal variability of the IOD amplitude is primarily caused by the remote influence of ENSO, while the forced weakening of the IOD amplitude is mainly induced by the suppression of local feedback processes in the tropical Indian Ocean.

## Remote ENSO influence on the IOD change

The quantification by the simple IOD model experiments shows that the remote ENSO forcing is the dominant source of the internal variability of the IOD amplitude, while it has a very small contribution

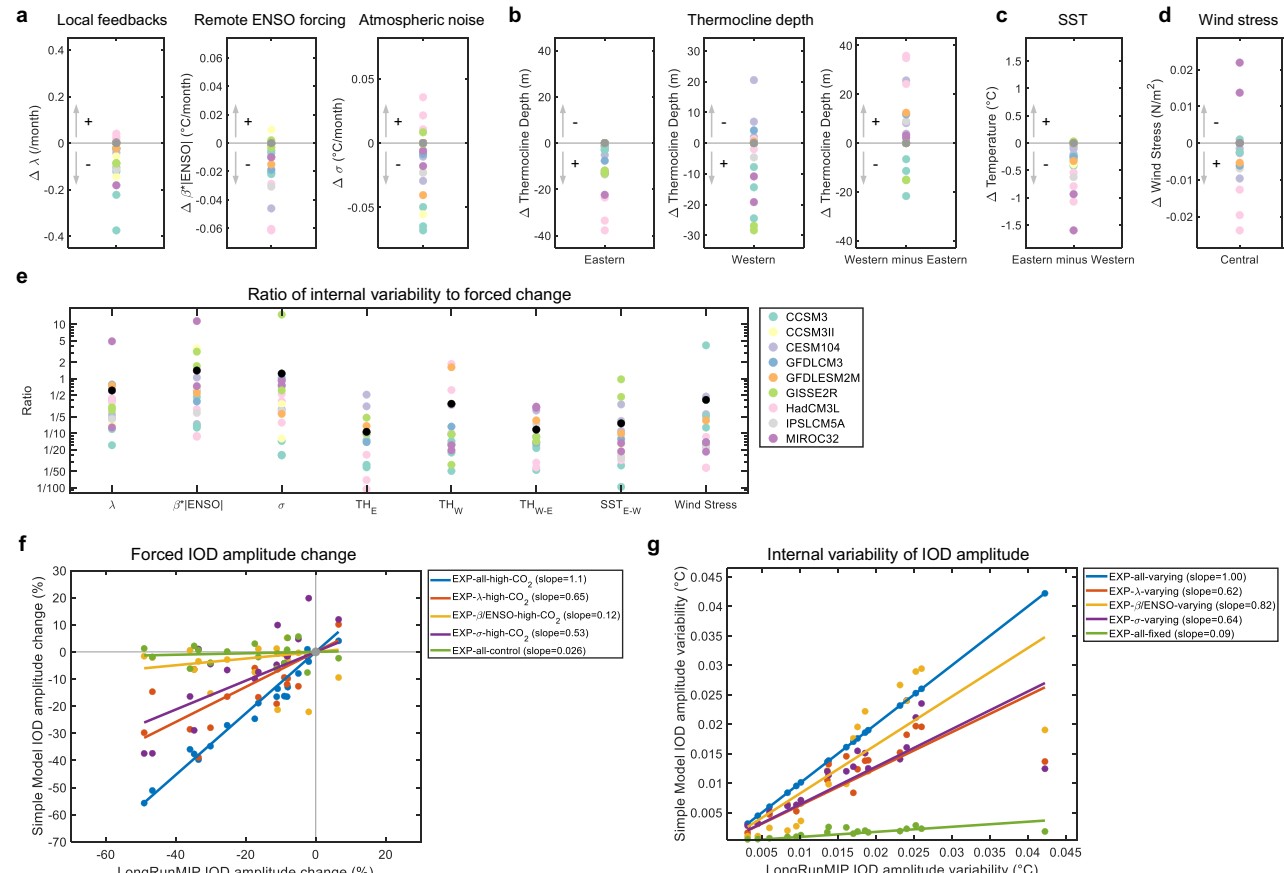

**Fig. 3 | Physical sources for the forced change and internal variability of Indian Ocean Dipole (IOD) amplitude. a–d** Changes in simple IOD model parameters and mean climatological states of the tropical Indian Ocean in the equilibrium period of high-$CO_2$ simulation. The change is calculated by subtracting the control simulation value from the high-$CO_2$ simulation value (colored dots). The plus (minus) sign indicates whether the change leads to an increase (decrease) in IOD amplitude (see Supplementary Discussion 3). **a** The changes in the annual mean $\lambda$ (left panel), $\beta \times ENSO$ (middle panel), and $\sigma$ (right panel). $\beta \times ENSO$ is a multiplication of the annual mean $\beta$ and El Niño–Southern Oscillation (ENSO) amplitude. **b** Changes in the east (90° E–110° E and 10° S–0° N) and west (50° E–70° E and 10° S–10° N) (middle panel) thermocline depths. Changes in the west minus east thermocline depth gradient (right panel). **c** Changes in the east (90° E–110° E and 10° S–0° N) minus west (50° E–70° E and 10° S–10° N) sea surface temperature (SST). **d** Changes in the central (70° E–90° E and 5° S–5° N) zonal wind stress. **e** The

ratio of internal variability to forced changes for the simple IOD model parameters and mean climatological states of the tropical Indian Ocean (same variables as in **a–d**). The standard deviation of the anomaly (deviation from a quadratic trend) is divided by the forced equilibrium change (deviation from the control experiment). A larger ratio indicates greater internal variability compared to the forced response. The black dot indicates the ensemble mean. **f** Results of the parameter perturbation experiments ("Methods" section). The original (x-axis) and reproduced IOD amplitude by the simple IOD model (y-axis). The percentage change in IOD amplitude from the control experiment is shown. The colored dots are the results of each high-$CO_2$ simulation. The linear regression line (without intercept) of the reproduced IOD amplitude versus the original IOD amplitude (solid line). The slope of the regression line is shown in the legend. **g** Results of the low-pass parameter experiment ("Methods" section). Same as (**f**), but the standard deviation of IOD amplitude variation (i.e., the strength of internal variability) are compared.

to the forced IOD amplitude change. This contrasting role of the ENSO forcing can be understood in physical terms.

The remote ENSO forcing on the IOD occurs by altering the atmospheric circulation in the tropical Indian Ocean basin[17,31,34,35]. The example ENSO regression pattern for wind stress and SST anomaly in the tropical Indian Ocean is shown in Supplementary Fig. 17. The warm (cold) phase of ENSO induces an anomalous easterly (westerly) wind stress in the eastern and central tropical Indian Ocean. This anomalous easterly (westerly) wind stress strengthens the wind-thermocline-SST feedback which acts as positive feedback, and consequently facilitates the development of the positive (negative) phase of the IOD. Thus, the ENSO-induced IOD variability is linked to the wind stress variability in the eastern and central tropical Indian Ocean.

In the high-$CO_2$ simulations, the eastern and central wind stress variability in the tropical Indian Ocean is robustly weakened (Supplementary Fig. 18). This forced weakening of the wind stress variability is correlated with the decrease in the remote ENSO forcing (Supplementary Fig. 18) and also with the ENSO-induced decrease in IOD amplitude (Supplementary Fig. 19). This suggests that the forced

decrease in ENSO forcing weakens the wind stress variability in the tropical Indian Ocean and has a subsequent effect on the decrease in IOD amplitude. However, the reduced ENSO forcing actually has a very small contribution to the decrease in IOD amplitude as shown in the simple IOD model experiments. This is because the IOD feedback processes are related not only to the wind stress variability, but also to multiple mean states of the tropical Indian Ocean, including thermocline depth, zonal SST gradient, and surface-subsurface coupling strength (e.g., sensitivity of SST to thermocline depth perturbation) (Supplementary Discussion 3). The forced change in these multiple factors would also contribute to the change in the feedback process and the associated IOD amplitude. Thus, the ENSO-induced weakening of the wind stress variability and associated wind-thermocline-SST feedback alone has a limited impact on the net feedback process, and is therefore unlikely to induce a significant change in IOD amplitude.

In contrast, the eastern and central wind stress variability can considerably contribute to the internal variability of IOD amplitude. The eastern and central wind stress variability shows a very high level of internal variability (Supplementary Fig. 20). The internal variability

of eastern and central wind stress is associated with the internal variability of ENSO and its remote forcing on IOD (Supplementary Fig. 21), and is closely linked with the internal variability of IOD amplitude (Supplementary Figs. 22 and 23). This suggests that the internal variability of ENSO influences the eastern and central wind stress variability in the tropical Indian Ocean, and contributes to the internal variability of IOD amplitude. The strong internal variability of ENSO and the associated remote forcing by the anomalous wind stress variability make the ENSO forcing the largest contributor to the internal variability of IOD amplitude.

## Discussion

Our results provide one explanation why the CMIP5/6 future projections show a widespread response of the IOD amplitude despite the presence of a robust warming pattern in the tropical Indian Ocean, in terms of the masking effect of internal variability. In high-$CO_2$ simulations, the tropical Indian Ocean shows a robust forced change in mean climatological states; the eastern and western thermocline depths decrease (Fig. 3b), the zonal SST gradient decreases (i.e., positive IOD-like change) (Fig. 3c), and the central Indian Ocean wind stress shows anomalous easterlies (Fig. 3d). Following the changes in the mean climatological state and the associated suppression of local feedback processes, the IOD amplitude robustly decreases in the warm equilibrium climate. This indicates that the IOD amplitude is tightly correlated to the mean state of the tropical Indian Ocean under an equilibrium climate condition, consistent with the conventional notions on the IOD dynamics. In contrast, under a transiently warming climate, the IOD amplitude is heavily influenced by internal variability attributed to the remote ENSO influence, and this internal variability obscures the relationship between the mean state of the tropical Indian Ocean and the IOD amplitude. Consequently, the internal variability weakens the correlation between the IOD amplitude and the mean climatological state of the tropical Indian Ocean.

Our results show that anthropogenic climate warming leads to a suppression of the IOD variability in the long term, offering a compelling argument for a human influence on IOD variability change with a high degree of inter-model consensus. The long-term climate simulations allow us to disentangle the forced response and internal variability, and to clarify the forced response of IOD to greenhouse warming. Although the detailed warming pathways differ from the typical CMIP5/6 future projection, our results clearly reveal the physics of the IOD response to global warming. It implies that the absence of clear anthropogenic signals in the historical and future IOD variability[21–23] can be partly attributed to the strong masking effect of internal variability which includes remote ENSO influence. Furthermore, our study suggests the possibility of dramatic changes in the characteristics of IOD in a very warm climate environment, such as the occurrence of double peaks in the boreal spring and fall, and a zonally uniform oscillation pattern, as seen in the CESM1.0.4 simulations. Given the widespread influence of IOD on the global hydrological patterns[6,10–13], such changes would likely cause shifts in monsoon precipitation and affect hydrological extremes and wildfire occurrences in Indian Ocean-rim countries.

## Methods
### LongRunMIP data

We use simulation output from the LongRunMIP archive. The Long-RunMIP archive includes a total of 11 models that provide monthly surface temperature (or monthly surface air temperature) variables longer than 500 years. We evaluate the IOD simulation performance of the 11 models, and select 9 models that can reasonably simulate the observed spatial and temporal characteristics of IOD. The model evaluation process and selection criteria are explained in Supplementary Discussion 1. The selected models are listed Supplementary Table 1.

The selected analysis sample includes a total of 27 simulations from 9 models. The simulation sample consists of 9 control and 18 high-$CO_2$ simulations (each model has a single number of control experiments). The control simulation is preindustrial run with a constant 1850 forcing. The high-$CO_2$ simulation includes the following forcing scenarios: instantaneous doubling (abrupt2x), quadrupling (abrupt4x), and octupling (abrupt8x); and gradual increase of 1% per year until doubling (1pct2x) and quadrupling (1pct4x) of atmospheric $CO_2$ levels relative to the control simulation. As an exception, CCSM3II includes the abrupt700ppm and abrupt1400ppm experiments. These experiments abruptly increased $CO_2$ levels to 700 ppm and 1400 ppm from the control level, respectively. After the increase in the $CO_2$ level, the $CO_2$ level remains unchanged until the end of the simulation. The length of the simulation differs by simulation (Supplementary Table 1), but all simulations are longer than 500 years. The analysis sample of the high-$CO_2$ simulations does not include their control simulation period. Details of the LongRunMIP protocol can be found in ref. 27.

Monthly surface temperature (variable name: ts) is used as a proxy for the SST since monthly SST is not consistently available (only a few models offer monthly SST data). Ref. 28 also applied a similar method to study ENSO amplitude changes using the LongRunMIP archive. For GISS-E2-R, monthly surface air temperature (variable name: tas) is utilized instead of monthly surface temperature, which is unavailable.

We use annual ocean potential temperature (variable name: thetao) and zonal wind stress (variable name: tauuo) variables. The monthly ocean potential temperature and zonal wind stress are not consistently available (only a few models offer monthly data). The ocean potential temperature is used for the calculation of the thermocline depth. The ocean potential temperature is unavailable for the CCSM3II model, and the zonal wind stress is unavailable for the CCSM3II and GISS-E2-R models. Therefore, these models are excluded from the results related to the thermocline depth and zonal wind stress.

### CMIP6 data

We use the piControl experiment from the CMIP6 archive[36]. A total of 36 models are used. The model name, ensemble, and simulation length are listed in Supplementary Table 2. We use monthly SST (variable name: tos) for the DMI calculation.

### Calculation of the DMI and Niño 3.4 index

The DMI is calculated for the LongRunMIP and CMIP6 simulation output. For the LongRunMIP, we applied quadratic detrending to the monthly SST to remove the forced long-term change (i.e., secular mode). The detrending is performed for the entire period of the simulation (see Supplementary Table 1 for the length of each simulation). The anomaly is defined as a deviation from this long-term trend. The area-averaged monthly SST anomaly is calculated for the western (50° E–70° E and 10° S–10° N) and eastern (90° E–110° E and 10° S–0° N) tropical Indian Ocean. The DMI is calculated as the west minus east monthly SST anomaly. The Niño 3.4 index is calculated as the area-averaged monthly SST anomaly for the Niño 3.4 region (170° W–120° W and 5° S–5° N). As a result, for each simulation, we obtain the DMI and Niño 3.4 index time series, the length of which is equal to the length of the entire simulation. Using these DMI and Niño 3.4 index time series, we perform analyses including the calculation of IOD amplitude and event intensity.

### IOD and ENSO amplitude

The amplitude of IOD and ENSO is defined as the standard deviation of the DMI and Niño 3.4 index, respectively.

### Detection of IOD events

The positive (negative) IOD event is defined as a three-month running mean of DMI exceeding (below) the one standard deviation of the time

series for at least three consecutive months. The intensity of a positive (negative) IOD event is the highest (lowest) DMI level during the event.

## Definition of the equilibrium and transient response

The equilibrium IOD response is defined as the response of IOD variability to an increased equilibrium GMST level. The equilibrium response reflects the forced equilibrium response due to greenhouse gas warming. The transient response is defined as the evolution of IOD variability in response to increasing GMST with time. The transient response reflects both the forced response and internal variability. Specifically, we interpret the long-term trend in IOD amplitude (or IOD event intensity) as the forced response, and the fluctuation in IOD amplitude (or IOD event intensity) that deviates from the long-term forced trend as the internal variability.

## Spectrum analysis

The spectrum analysis is performed using the Multi-Taper Method (MTM), a widely used technique for spectrum estimation. The parameters of the MTM analysis are chosen by following the standard guidelines of ref. 37,38. We set the time-frequency bandwidth parameter ($NW$) to 2, and the number of Slepian Tapers ($K$) to 3. The significance of the power spectrum is estimated based on the red noise, which is a typical null hypothesis for climate variability. The 99%, 95%, and 50% significance levels are calculated using the Monte Carlo method with 100 ensembles.

## Equilibrium sensitivity

We calculate the equilibrium sensitivity of the IOD amplitude to GMST. We use the IOD amplitude and GMST in the equilibrium period (i.e., the last 500 years of the simulation). For each model, we calculate the linear regression coefficients of $\{DMI\}_s = a_{eq}\{GMST\}_s + b_{eq}$ where $\{DMI\}_s$ and $\{GMST\}_s$ are an array of IOD amplitude and GMST for the control and high-$CO_2$ simulations, respectively. The $a_{eq}$ is the equilibrium sensitivity. For $\{DMI\}_s$, we use percentage change from the control simulation, not the absolute value. For example, in the equilibrium period, the CCSM3 model exhibits IOD amplitudes of 0.26 °C, 0.17 °C, 0.14 °C, and 0.13 °C for the control, abrupt2x, abrupt4x, and abrupt8x simulations, respectively. The corresponding percentage changes in the IOD amplitude from the control simulations are 0%, −34%, −47%, and −50%, respectively. The corresponding GMSTs are 14.3 °C, 16.6 °C, 18.9 °C, and 21.5 °C, respectively. We perform linear regression on this array of IOD amplitude ($\{DMI\}_s = \{0\%, -34\%, -47\%, -50\%\}$) and GMST ($\{GMST\}_s = \{14.3 °C, 16.6 °C, 18.9 °C, 21.5 °C\}$) and estimate the $a_{eq}$. The unit of $a_{eq}$ is %/°C. This equilibrium sensitivity quantifies the percentage change in IOD amplitude per 1 °C increase in GMST under an equilibrium climate condition.

## Transient sensitivity

We calculate the transient sensitivity of IOD amplitude to GMST. We use time series of 100-year moving IOD amplitude and GMST. For each high-$CO_2$ simulation, we perform moving window linear regression for $\{DMI\}_w = a_{ts}\{GMST\}_w + b_{ts}$ where $\{DMI\}_w$ and $\{GMST\}_w$ are times series of 100-year moving IOD amplitude and GMST. The $a_{ts}$ is the transient sensitivity. For $\{DMI\}_w$, we use percentage change from the initial value, not the absolute value. The unit of $a_{ts}$ is %/°C. The size of moving window is 1 °C. The statistics of the $a_{ts}$ distribution (mean and standard deviation) is presented in Fig. 2b. The full probability distribution of $a_{ts}$ is presented in Supplementary Fig. 8. To derive the probability distribution of the $a_{ts}$, we perform kernel density estimation on the $a_{ts}$. The bandwidth is set as the optimal value for normal densities.

## Calculation of thermocline depth

The thermocline depth is calculated as the maximum gradient depth of the ocean potential temperature profile. For each horizontal grid point, we perform a cubic interpolation on the vertical ocean potential

temperature profile. The maximum temperature gradient depth is determined in the interpolated ocean potential temperature profile.

## Simple IOD model

The simple IOD model[20,31] is written as

$$\frac{dT}{dt} = \lambda T + \beta ENSO(t) + \sigma\xi \qquad (1)$$

where $T$ is the DMI and $ENSO$ is the Niño 3.4 index. The simple IOD model includes three terms representing local feedback processes ($\lambda T$), external ENSO forcing ($\beta ENSO(t)$), and stochastic forcing ($\sigma\xi$). $\lambda$ characterizes local feedback processes associated with the IOD. $\beta$ characterizes the sensitivity of the DMI to external ENSO forcing. $\sigma$ is the noise amplitude and $\xi$ is white noise with zero mean and unit standard deviation. We consider the seasonality of $\lambda$, $\beta$, and $\sigma$. Thus, these three parameters vary seasonally. The model can consider the lag between ENSO and $T$ because the model equation relates the ENSO with the tendency of $T$, not with the $T$ itself.

We note that there are several versions of the simple IOD model in the current literature depending on the types of feedback processes included. The simple IOD model used in this study is the basic type model introduced in ref. 31, which considers the essential physical process of IOD in the simplest way. Hence, the model does not explicitly account for the time-delayed Rossby wave process as done in ref. 20, and implicitly considers it by $\lambda T$, which comprehensively represents local feedback processes in the tropical Indian Ocean. The simplicity of the model used in this study has advantages that facilitate the physical understanding of the model results, especially in determining whether the IOD amplitude change is caused by a local process or a remote influence.

## Fitting the parameters of the simple IOD model

We perform parameter fitting of the simple IOD model for the given monthly DMI and Niño 3.4 index time series. We estimate the values of three parameters ($\lambda$, $\beta$, $\sigma$) for each month of the year from January to December. We discretize the simple IOD model as follows:

$$\frac{\Delta T_i}{\Delta t} = \lambda T_i + \beta(EMSO)_i \qquad (2)$$

where $i$ denote time step ($i = 1, 2...N$) and $\Delta T_i = T_{i+1} - T_i$. $\Delta t$ is 1 month. We perform the linear regression and estimate $\lambda$ and $\beta$ using yearly time series of $\Delta T_i$, $T_i$, and $(ENSO)_i$ for each month of the year. The $\sigma$ is estimated as the standard deviation of the residual of the linear regression.

## Solving the simple IOD model

The simple IOD model is numerically solved using the Euler−Maruyama method with a time step of 0.1 months. The Euler−Huen method is a numerical scheme for solving the Stochastic Differential Equation. The input data is seasonally varying parameter and the Niño 3.4 index. We performed the linear interpolation for the parameters and the Niño 3.4 index, which have a time resolution of 1.0 month, and downscaled the time resolution to 0.1 month to match the time resolution with the numerical time step.

## Parameter perturbation experiment

We perform the parameter perturbation experiment using the simple IOD model. The parameters of the simple IOD model are fitted using the DMI and Niño 3.4 index time series in the equilibrium period. The fitted parameters are shown in Supplementary Fig. 11 and Fig. 3a. The statistical relationships between the fitted parameters and the GMST are shown in Supplementary Fig. 24. We reproduce the DMI time series by running the simple IOD model using the fitted parameters (EXP-*all*-

*high-CO₂*) for each LongRunMIP simulation. We then run the simple model with the control simulation parameters but with the replaced $\lambda$ (EXP-*λ-high-CO₂*), $\beta$ (EXP-*β-high-CO₂*), ENSO (EXP-ENSO-*high-CO₂*), $\beta$, and ENSO (EXP-*β*ENSO-*high-CO₂*), and $\sigma$ (EXP-*σ-high-CO₂*) values with the ones from the high-CO₂ simulations. For example, the EXP-*λ-high-CO₂* uses $\lambda$ from the high-CO₂ simulation and the $\beta$, ENSO, and $\sigma$ from its control simulation. The parameter perturbation experiment is performed for each high-CO₂ simulation. Additionally, for the baseline experiment, we run the simple model using the control simulation parameters (EXP-*all-control*).

### Low-pass parameter experiment

We perform the low-pass parameter experiment using the simple IOD model. The model parameters are fitted for a moving 100-year window using the DMI and Niño 3.4 index time series. The annual mean value of the fitted parameters is shown in Supplementary Figs. 12–14 (note that the parameter fitting is performed for each calendar month as shown in Supplementary Fig. 11, but we only present their annual mean value for simplicity). For each segment of a window, we reproduced the DMI time series using the fitted parameters and calculated IOD amplitude (EXP-*all-varying*). To remove the fluctuations of the parameters, we low-pass filter $\lambda$, $\beta$ENSO, and $\sigma$ and only leave their quadratic trend. We run the EXP-*all-varying* but with the replaced original parameter from the low-pass filtered parameters for– $\lambda$ (EXP-*λ-varying*), $\beta$ (EXP-*β-varying*), ENSO (EXP-ENSO-*varying*), $\beta$ and ENSO (EXP-*β*ENSO-*varying*), and $\sigma$ (EXP-*σ-varying*). For example, the EXP- *λ-varying* uses the original $\lambda$ and low-pass filtered $\beta$, ENSO, and $\sigma$. The strength of the internal variability of IOD amplitude is quantified as the standard deviation of the IOD amplitude. The low-pass parameter experiment is performed for each high-CO₂ simulation. Additionally, for the baseline experiment, we run the simple model solely using only the low-pass parameters (EXP-*all-fixed*).

## Data availability

The LongRunMIP data are available upon request to Maria Rugenstein (maria.rugenstein@colostate.edu). The CMIP6 data are available from the Earth System Grid Federation repository (https://esgf-node.llnl. gov/projects/cmip6/). ERSSTv5 is available from https://psl.noaa.gov/ data/gridded/data.noaa.ersst.v5.html. COBE is available from https:// psl.noaa.gov/data/gridded/data.cobe.html. HadISST is available from https://www.metoffice.gov.uk/hadobs/hadisst/.

## Code availability

The codes used in this study are available from the authors upon request.

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

## Acknowledgements
This work was supported by a National Research Foundation of Korea (NRF) grant funded by the Korean government (MSIT) (NRF-2018R1A5A1024958) (S.-I.A.) and the Yonsei Signature Research Cluster Program of 2021 (2021-22-0003) (S.-I.A.).

## Author contributions
S.-K.K. conceived the main idea of the research and wrote the draft. H.-J.P. produced the initial analysis results of the LongRunMIP IOD amplitude change. S.-K.K. performed the main analysis/experiments and produced figures. H.-J.P. and S.-I.A. assisted in the main analysis. S.-K.K., H.-J.P., S.-I.A., C.L., W.C, A.S. and J.-S.K. discussed the results and wrote the manuscript.

## Competing interests
The authors declare no competing interests.
