## [Peer Review File · Nature Communications]

Decreased Indian Ocean Dipole variability under prolonged greenhouse warmingREVIEWER COMMENTS

Reviewer #1 (Remarks to the Author):

The authors investigated the variability of IOD amplitude under prolonged greenhouse warming based on model simulations, and found that IOD variability weakens under prolonged greenhouse warming due to the suppressed local feedback processes in tropical Indian Ocean. On centennial time scale, the weakening trend is disturbed by the internal variability associated with ENSO. Although these results are interesting and show the possible response of IOD to global warming on millennial-length scale, some major conclusions are not convincing and further evidences on physical processes are needed.

1 Since the conclusions are obtained based on model results, the capability of models in reproducing IOD needs to be carefully verified. However, the models are not validated and selected before they were used for analysis in the manuscript. This may affect the credibility of the results. The authors should select the models based on the fidelity of model results on the temporal and spatial characteristics of IOD. Objective criteria are needed for model selection.

2 The first EOF mode of the surface temperature anomaly over tropical Indian Ocean in CESM1.0.4 simulations (Supplementary Fig. 7) seems to be different from that in observations. In observations, IOD accounts for the second EOF mode rather than the first EOF mode. In addition, the SST variation is strongest in the off-Sumatra region in observations, while in simulations there are two regions with strong SST variations. This may deserve further discussion.

3 Some major conclusions rely completely on model results but are not well supported by physical mechanisms. The authors concluded that (L221) internal variability of the IOD amplitude is primarily caused by the remote influence of ENSO, while the forced weakening of the IOD amplitude is mainly induced by the suppression of local feedback processes in tropical Indian Ocean. Why is ENSO important on centennial time scale but not important on millennial time scale? What are the underlying mechanisms?

(L216) The internal variability in $\beta \times \text{ENSO}$ is the most dominant factor for the internal variability of IOD amplitude. How does ENSO influence the IOD amplitude? Is it due to variations in ENSO intensity or the variations in the connection between ENSO and IOD?

4 The authors suggest that, in the high-CO₂ simulations, the tropical Indian Ocean shows a positive IOD-like change which weakens the local feedback processes. (Supplementary Discussion 2). But previous studies show that such IOD-like pattern could enhance the thermocline–SST feedback. What is the cause for such difference? Bjerknes feedback is the major feedback process in IOD development. Are the variations of the three components of Bjerknes feedback consistent with the conclusion?

5 The consensus of model results needs to be tested. The statistical significance needs to be ascertained. e.g. L252 Furthermore, our study suggests the possibility of dramatic changes in the characteristics of IOD in a very warm climate environment, such as the occurrence of double peaks in the boreal spring

and fall, and a zonally uniform oscillation pattern.
Are these results significant?

6 The exact meaning of some items, e.g. IOD amplitude, IOD variability, internal variability, transient IOD response, need to be further clarified, make them easier for readers to understand.

Reviewer #2 (Remarks to the Author):

Review for submission "Decreased Indian Ocean Dipole variability under prolonged greenhouse warming"

This study utilized LongrunMIP to address the forced long-term changes in the Indian Ocean Dipole (IOD) amplitude arising from natural variability. The manuscript is well-written, and the arguments are supported. Nevertheless, there are some concerns that the authors should address in the revision.

1. About the natural variability. Line 80: "IOD amplitude in the twenty-first century either increases or decreases depending on the pathways of internal variability, despite the same warming forcing." The term 'pathways' may be more closely associated with forcing rather than natural variability. Please consider revising for clarity."

2. About the simple IOD model. In paper 20, four terms are discussed: local feedback, delayed feedback, ENSO impact, and stochastic forcing. In this study, the authors only consider three terms, excluding the delayed feedback. Please provide more discussions or explanations on this consideration. Line 381: "We consider the seasonality of λ , β , and σ . Thus, these three parameters vary seasonally. The typical value of the annual mean λ , estimated from observations and climate models in previous studies, is negative.' It is assumed here that the model is fitted for each month with ENSO and DMI index in phase (lag-0). Whether you consider a lead-lag relationship? Also, for the annual mean model, does the equation still hold?"

3. About the seasonal changes. Paper 19 addressed that the early IOD is more frequently observed under warming conditions. In this study, Line 124 mentions, 'The seasonal variability of IOD decreases in almost all seasons (Supplementary Fig. 5).' Moreover, Figure S5 illustrates that HadCM3L and a few other models peak early in JJA. It would be beneficial to delve into a more detailed discussion of this change. Lines 135-146: You might consider separating the discussions on seasonal changes and spatial patterns into two paragraphs for clarity.

4. About the relationship between changes in IOD and ENSO. Paper 28 addressed long-term ENSO changes based on LongrunMIP. It would be valuable to explore whether models agree on changes in both IOD and ENSO and if the dominating mechanisms are similar. Authors should engage in discussions on this aspect.

5. About the definition of IOD amplitude. Line 102: 'We measure the IOD amplitude as the standard deviation of the DMI.' Typically, the community uses the peak season (SON) DMI to gauge IOD changes. Here, monthly DMI is employed; the authors may wish to explain this choice.

6. About transient response. Line 95: For the 1pct2x and 1pct4x runs, 'After the CO₂ increase, the forcing is stabilized for typically longer than 1,000 years.' While the first 200 years may be considered the transient period, the runs predominantly reflect the equilibrated response after the forcing stabilizes. Line 148: 'We analyze the transient response of the IOD amplitude to the CO₂ forcing. We calculate the 100-year moving IOD amplitude for the entire simulation period.' It is noted that the entire period (1000 years or more) covers both the transient and equilibrated periods. The authors may need to consider taking the first 100-200 years as the transient period when the forcing is changing. After the forcing stabilizes, it may not be considered part of the transient period.

7. About the period of analysis. Line 107: 'last 500 years.' Are the entire 500 years used for detrending through quadratic fitting? Line 312: 'For the LongRunMIP, we applied quadratic detrending to the monthly SST to remove the forced long-term change (i.e., secular mode).' Could the authors clarify whether they used a 100-year or 500-year period for this detrending process?

8. Figure S13. The upper panel x-y axis should be swapped to be consistent with the lower panel.

Author's Reply to the Reviewer #1

Reviewer #1 (Remarks to the Author):

The authors investigated the variability of IOD amplitude under prolonged greenhouse warming based on model simulations, and found that IOD variability weakens under prolonged greenhouse warming due to the suppressed local feedback processes in tropical Indian Ocean. On centennial time scale, the weakening trend is disturbed by the internal variability associated with ENSO. Although these results are interesting and show the possible response of IOD to global warming on millennial-length scale, some major conclusions are not convincing and further evidences on physical processes are needed.

Reply: We appreciate your recognition and valuable comments on our paper. Thanks to your constructive comments, we have made a substantial improvement to the manuscript. The revised manuscript contains some major updates as follows:

- 1.** We evaluated the performance of the LongRunMIP model and selected models that can reasonably simulate the IOD. We updated the analyses in the paper with these selected models (9 out of 11 models). For the model selection process and criteria, please see the newly added section **Supplementary Discussion 1**.
- 2.** We performed the physical mechanism analysis on how ENSO influences IOD. Please see the newly added section **Remote ENSO influence on the IOD change**.

The reply to your comment is attached below. Please note that we will refer to the text line and figure/table number based on the revised version of the manuscript.

1 Since the conclusions are obtained based on model results, the capability of models in reproducing IOD needs to be carefully verified. However, the models are not validated and selected before they were used for analysis in the manuscript. This may affect the credibility of the results. The authors should select the models based on the fidelity of model results on the temporal and spatial characteristics of IOD. Objective criteria are needed for model selection.

Reply: In the original manuscript, we used all 11 available LongRunMIP models for the IOD analysis. Following your comment, we performed a model performance evaluation and identified the model that can reasonably simulate IOD. We selected 9 out of 11 models that can reasonably simulate the observed characteristics of IOD and used them for the analysis.

The detailed model evaluation and selection process is fully documented in **Supplementary Discussion 1**. We evaluate model performance based on the DMI, the main analysis variable of this study. We examine three basic characteristics of the IOD: (1) the peak season of the DMI standard deviation, (2) the DMI skewness sign, and (3) the spatial regression pattern of the DMI. The peak DMI season is related to the seasonal phase locking of the IOD, which is associated with various physical processes including the local feedback process and ENSO. Therefore, it is a good test metric for whether the model correctly simulates the IOD feedback process and associated variability. The DMI skewness characterizes the nonlinearity of the IOD, and is related to the response of the IOD to greenhouse gas warming. The DMI spatial regression pattern shows the spatial pattern of the IOD. We evaluate the ability of the model to simulate these three metrics. Based on the evaluation, we score the model performance level and classify it into four classes: high, medium, low, and very low. The summary of the model evaluation is shown in **Table R1**.

We select the high and medium performance models for the IOD analysis, and exclude the low performance models. Finally, out of 11 models, the following 9 models are selected: CCSM3, CCSM3II, CESM1.0.4, GFDL-CM3, GFDL-ESM2M, GISS-

E2-R, HadCM3L, IPSL-CM5A and MIROC3.2. The following 2 models are excluded from the analysis: CNRMCM61 and MPIESM12.

We accordingly updated all related material in the manuscript. First, we removed the results of CNRMCM61 and MPIESM12, and updated all related figures, tables, and metrics. Second, we added an explanation of the model evaluation and selection process. We newly added **Supplementary Discussion 1** with **Supplementary Figs. 25** and **26** and **Supplementary Table 3**. We also accordingly added explanations in **L100** and **L349**.

Our model selection process does not affect the main results of the paper, but rather strengthens them. **Table R2** shows the inter-model (or inter-simulation) consensus ratio of the IOD response for three different model groups, all models (11 models; model samples in the original manuscript), selected high and medium performance models (9 models; model samples in the revised manuscript), and selected high performance models (6 models). As you can see, the selected high and medium performance model group shows a higher inter-model consensus on the IOD response than the all models group. In particular, for the selected high performance model group, the inter-model consensus ratio reaches almost 1. This increasing inter-model consensus ratio adds confidence to the results of the paper. The main results of the parameter perturbation experiment (**Fig. 3f**) and the low-pass parameter experiment (**Fig. 3g**) which show the physical source of the IOD amplitude change, also remain almost unchanged. Therefore, the main results and the associated conclusion of the paper remain the same after applying the model selection.

Table R1. Model evaluation summary. Same as Supplementary Table 3.

Type	Product	Peak DMI Standard Deviation Season	DMI Skewness Sign	DMI Regression Pattern (Peak Positions)	Model Performance Level (Score)
Observation (1900-2022)	ERSSTv5	October	Positive	Dipole (Sumatra-Java coast / East African coast)	-
	COBE	October	Positive	Dipole (Sumatra-Java coast / East African coast)	-
	HadISST	September	Positive	Dipole (Sumatra-Java coast / East African coast)	-
LongRunMIP (control)	CCSM3	September	Positive	Dipole (Sumatra-Java coast / East African coast)	High (3)
	CCSM3II	September	Positive	Dipole (Sumatra-Java coast / East African coast)	High (3)
	CESM104	October	Positive	Dipole (Sumatra-Java coast / East African coast)	High (3)
	CNRMCM61	December	Negative	Dipole (Sumatra-Java coast / East African coast)	Low (1)
	GFDLCM3	September	Positive	Dipole (Sumatra-Java coast / East African coast)	High (3)
	GFDLESM2M	September	Negative	Dipole (Sumatra-Java coast / East African coast)	Medium (2)
	GISSE2R	October	Positive	Dipole (Sumatra-Java coast / Arabian Sea)	Medium (2)
	HadCM3L	October	Positive	Dipole (Sumatra-Java coast / Arabian Sea)	Medium (2)
	IPSLCM5A	October	Positive	Dipole (Sumatra-Java coast / East African coast)	High (3)
	MIROC32	September	Positive	Dipole (Sumatra-Java coast / East African coast)	High (3)
MPIESM12	August	Negative	Dipole (Sumatra-Java coast / East African coast)	Low (1)	

Table R2. The inter-model (or inter-simulation) consensus ratio of the IOD response for different model groups. The consensus ratio for all models (11 models; model samples in the original manuscript), selected high and medium performance models (9 models; model samples in the revised manuscript), and selected high performance model (6 models).

Response Type	Variable	Response in the Majority	All Model	Selected Model (High and Medium Performance)	Selected Model (High Performance)
Equilibrium Response	IOD Amplitude	Decrease	17 out of 23 (74%)	15 out of 18 (83%)	11 out of 12 (92%)
	IOD Event Intensity (Positive)	Decrease	20 out of 23 (87%)	18 out of 18 (100%)	12 out of 12 (100%)
	IOD Event Intensity (Negative)	Decrease	17 out of 23 (74%)	15 out of 18 (83%)	11 out of 12 (92%)
	IOD Amplitude Sensitivity	Negative	9 out of 11 (82%)	8 out of 9 (89%)	6 out of 6 (100%)
Transient Response	IOD Amplitude Sensitivity (Mean)	Negative	21 out of 23 (91%)	17 out of 18 (94%)	12 out of 12 (100%)
	IOD Amplitude Sensitivity (Long-term Trend)	Negative	20 out of 23 (87%)	17 out of 18 (94%)	12 out of 12 (100%)

2 The first EOF mode of the surface temperature anomaly over tropical Indian Ocean in CESM1.0.4 simulations (Supplementary Fig. 7) seems to be different from that in observations. In observations, IOD accounts for the second EOF mode rather than the first EOF mode. In addition, the SST variation is strongest in the off-Sumatra region in observations, while in simulations there are two regions with strong SST variations. This may deserve further discussion.

Reply: Following your comment, we carefully reexamined the EOF analysis results of the CESM1.0.4 simulations.

(1) The first and second EOF mode

As we explain in the Methods section, we use surface temperature (variable name: ts) as a proxy for the SST (variable name: tos) because monthly SST is not consistently available (only a few models provide monthly SST data). It has come to our attention that we did not apply the land masking to the surface temperature variable when performing the EOF analysis on it. Thus, the original EOF analysis results included both land and sea surface temperature anomalies (we applied the land masking for all other calculations, but the CESM1.0.4 EOF analysis was the only exception).

To correct this, we applied the land masking to the surface temperature field and re-performed the EOF analysis. Please see **Fig. R1**.

Fig. R1. EOF analysis of the CESM1.0.4 surface temperature anomaly. The first and second EOF modes of the monthly surface temperature anomaly over the

tropical Indian Ocean (40° E-120° E and 20° S-20° N). The results for the control (a), abrupt2x (b), abrupt4x (c), and abrupt8x (d). The upper panel is the first mode and the lower panel is the second mode. Variance explained by the mode is marked in the title of each panel.

As you can see, the IOD now accounts for the second EOF mode, consistent with the observation.

We accordingly revised the figure and its caption with the land mask applied result shown in **Fig. R1**. Please see the updated **Supplementary Fig. 7**. We slightly extended the eastern edge of the analysis domain from 110° E to 120° E to more clearly show and discuss the secondary strong SST location, that you pointed out in your comment. We also removed the EOF analysis result for the SON surface temperature anomaly for a fair comparison between the simulations, as the seasonal phase locking pattern is significantly different in the abrupt8x (secondary seasonal peak emerges in March-April-May).

(2) Difference in the spatial pattern

As you pointed out, there is a difference in the EOF pattern between the CESM and the observation. We added a discussion of this difference in the caption of **Supplementary Fig. 7** as (we also considered adding it in the main text, but given the flow of the paper, we decided to add it in the figure caption):

"We note that the spatial pattern of the second EOF mode of the CESM1.0.4 shows some difference from the typical observation. Both the CESM1.0.4 and the historical observational records show a strong SST EOF pattern in the off-Sumatra region, but the CESM1.0.4 also shows a strong SST in the Sumatra coastal region. The difference can be partly attributed to the bias of the CESM1.0.4 in simulating the mean climatological state of the tropical Indian Ocean, including the wind stress field, the SST pattern, and the ocean potential temperature profile. A detailed discussion of this subject is beyond the scope of this study which focuses primarily

on the response of the tropical Indian Ocean to greenhouse gas warming. Despite such bias in the mean state, our conclusion in this paper is well supported by the strong inter-model consensus in the IOD response.”

Thank you.

3 Some major conclusions rely completely on model results but are not well supported by physical mechanisms. The authors concluded that (L221) internal variability of the IOD amplitude is primarily caused by the remote influence of ENSO, while the forced weakening of the IOD amplitude is mainly induced by the suppression of local feedback processes in tropical Indian Ocean. Why is ENSO important on centennial time scale but not important on millennial time scale? What are the underlying mechanisms?

(L216) The internal variability in $\beta \times \text{ENSO}$ is the most dominant factor for the internal variability of IOD amplitude. How does ENSO influence the IOD amplitude? Is it due to variations in ENSO intensity or the variations in the connection between ENSO and IOD?

Reply: Following your comment, we performed a series of additional experiments and analyses.

(1) Disentangling the role of ENSO intensity and IOD sensitivity to ENSO

In the original version of the paper, we quantified the contribution of remote ENSO forcing ($\beta \times \text{ENSO}$) to the forced change and internal variability of IOD amplitude. To separate the contribution of ENSO variability (ENSO) and the IOD sensitivity to ENSO variability (β), we performed additional experiments. We repeated the previous parameter perturbation and low-pass parameter experiment, but changed ENSO and β separately.

Please see the newly added experiment result in **L245** and associated **Supplementary Figs. 15** and **16**. The added explanation of the result is read as

follows:

“The remote ENSO forcing is a combined effect of ENSO variability (ENSO) and sensitivity of IOD to ENSO variability (β). We perform additional experiments to separately quantify their contribution to the forced change and the internal variability of the IOD amplitude (Methods). The changes in both ENSO and β make virtually no contribution to the forced decrease in IOD amplitude (Supplementary Fig. 15). The decrease in ENSO variability makes a larger contribution than β , but the contribution is very small compared to λ and σ . The internal variability of ENSO alone makes a considerable contribution to the internal variability of the IOD amplitude, similar in magnitude to that of λ and σ , and β makes a slightly smaller contribution (Supplementary Fig. 16). This shows that the large internal variability of $\beta \times \text{ENSO}$ and its influence on IOD is attributed to combined internal variability of ENSO and β .”

In answer to your question, the additional experiment shows that the internal variability of $\beta \times \text{ENSO}$ is caused by contribution from both internal variability of ENSO amplitude (i.e., the standard deviation of ENSO) and internal variability of the sensitivity of IOD to ENSO variability (β). The internal variability of ENSO amplitude is the primary contributor, and β is the secondary effect, however, they have a similar magnitude of contribution.

(2) Physical mechanism of the remote ENSO forcing

We performed the analysis on the physical mechanism of the remote ENSO forcing on the IOD amplitude change. Please see the newly added section, “**Remote ENSO influence on the IOD change**” in **L260** and associated **Supplementary Figs. 17-23**. The added section is read as follows.

“**Remote ENSO influence on the IOD change**”

The quantification by the simple IOD model experiments shows that the remote ENSO forcing is the dominant source of the internal variability of the IOD amplitude,

while it has a very small contribution to the forced IOD amplitude change. This contrasting role of the ENSO forcing can be understood in physical terms.

The remote ENSO forcing on the IOD occurs by altering the atmospheric circulation in the tropical Indian Ocean basin. The example ENSO regression pattern for wind stress and SST anomaly in the tropical Indian Ocean is shown in Supplementary Fig. 17. The warm (cold) phase of ENSO induces an anomalous easterly (westerly) wind stress in the eastern and central tropical Indian Ocean. This anomalous easterly (westerly) wind stress strengthens the wind-thermocline-SST feedback which acts as positive feedback, and consequently facilitates the development of the positive (negative) phase of the IOD. Thus, the ENSO-induced IOD variability is linked to the wind stress variability in the eastern and central tropical Indian Ocean.

In the high-CO₂ simulations, the eastern and central wind stress variability in the tropical Indian Ocean is robustly weakened (Supplementary Fig. 18). This forced weakening of the wind stress variability is correlated with the decrease in the remote ENSO forcing (Supplementary Fig. 18) and also with the ENSO-induced decrease in IOD amplitude (Supplementary Fig. 19). This suggests that the forced decrease in ENSO forcing weakens the wind stress variability in the tropical Indian Ocean and has a subsequent effect on the decrease in IOD amplitude. However, the reduced ENSO forcing actually has a very small contribution to the decrease in IOD amplitude as shown in the simple IOD model experiments. This is because the IOD feedback processes are related not only to the wind stress variability, but also to multiple mean states of the tropical Indian Ocean, including thermocline depth, zonal SST gradient, and surface-subsurface coupling strength (e.g., sensitivity of SST to thermocline depth perturbation) (Supplementary Discussion 3). The forced change in these multiple factors would also contribute to the change in the feedback process and the associated IOD amplitude. Thus, the ENSO-induced weakening of the wind stress variability and associated wind-thermocline-SST feedback alone has a limited impact on the net feedback process, and is therefore unlikely to induce a significant change in IOD amplitude.

In contrast, the eastern and central wind stress variability can considerably contribute to the internal variability of IOD amplitude. The eastern and central wind stress variability shows a very high level of internal variability (Supplementary Fig. 20). The internal variability of eastern and central wind stress is associated with the internal variability of ENSO and its remote forcing on IOD (Supplementary Fig. 21), and is closely linked with the internal variability of IOD amplitude (Supplementary Figs. 22 and 23). This suggests that the internal variability of ENSO influences the eastern and central wind stress variability in the tropical Indian Ocean, and contributes to the internal variability of IOD amplitude. The strong internal variability of ENSO and the associated remote forcing by the anomalous wind stress variability make the ENSO forcing the largest contributor to the internal variability of IOD amplitude.”

In addition to the newly added section, we would like to specifically answer your questions.

- How does ENSO influence the IOD amplitude?

The ENSO forcing influences the eastern and central wind stress in the tropical Indian Ocean by altering the Walker circulation pattern. This anomalous wind in the tropical Indian Ocean facilitates the development of IOD. Specifically, the warm (cold) phase of ENSO induces an anomalous easterly (westerly) wind stress in the eastern and central tropical Indian Ocean. This anomalous easterly (westerly) wind stress strengthens the wind-thermocline-SST feedback which acts as positive feedback, and facilitates the development of the positive (negative) phase of the IOD. Consequently, the ENSO forcing increases the IOD amplitude.

- Why is ENSO important on centennial time scale but not important on millennial time scale? What are the underlying mechanisms?

On the centennial time scale, internal variability dominates the change in IOD

amplitude as shown in the **“Transient IOD response”** section of the paper. The 100-year moving IOD amplitude fluctuates strongly with time under greenhouse gas warming. In contrast, on the millennial time scale, the forced change in IOD amplitude is detectable and shapes the IOD amplitude change. The forced change in IOD amplitude is consistently detectable with the long-term trend in the 100-year moving IOD amplitude (as shown in the **“Transient IOD response”** section) and the change in IOD amplitude under a different equilibrium warming level (as shown in the **“Equilibrium IOD response”** section). Therefore, the change in IOD amplitude over the centennial and millennial timescales can be translated as internal variability and forced change of IOD amplitude, respectively.

ENSO-induced IOD variability is linked to wind stress variability in the eastern and central tropical Indian Ocean. As we explained in the newly added section, the forced change in wind stress variability alone has a limited role in altering the net IOD feedback process, which involves with multiple physical factors. The decrease in wind stress variability possibly contributes to the decrease in IOD amplitude, but the series of analyses and experiments (explained in **“Physical sources of the IOD change”** and **“Remote ENSO influence on the IOD change”**) show that the magnitude is actually very small. Thus, ENSO does not have a significant influence on the forced IOD change that occurs on a millennial time scale.

In contrast, the wind stress variability shows a very strong degree of internal variability compared to other variables, which means that its centennial-scale temporal fluctuation is very large (we define internal variability as anomalous short-term fluctuation in the variable that deviates from the long-term forced change). The strength of the temporal fluctuation of the IOD amplitude is determined by the strength of the temporal fluctuation of the IOD feedback processes. Thus, the wind stress variability, which exhibits a considerable degree of fluctuation, can considerably contribute to the fluctuation in the IOD feedback processes and the IOD amplitude. Consequently, internal variability of ENSO forcing via the wind stress in the tropical Indian Ocean, contributes significantly to the internal variability of

IOD amplitude, that occurs on a centennial timescale. On this internal variability of ENSO forcing, the internal variability of ENSO amplitude is the primary contributor, and the internal variability of IOD sensitivity to ENSO is the secondary effect.

4 The authors suggest that, in the high-CO₂ simulations, the tropical Indian Ocean shows a positive IOD-like change which weakens the local feedback processes. (Supplementary Discussion 2). But previous studies show that such IOD-like pattern could enhance the thermocline–SST feedback. What is the cause for such difference? Bjerknes feedback is the major feedback process in IOD development. Are the variations of the three components of Bjerknes feedback consistent with the conclusion?

Reply: We assume that you are referring to the line “Conversely, the reduced east-to-west SST gradient (i.e., positive IOD-like change) (Fig. 3c) would weaken the surface zonal advective feedback” in **Supplementary Discussion 3** (previously Supplementary Discussion 2, but renumbered). Our intention was to discuss the individual effect of a reduced mean zonal SST gradient and associated zonal advective feedback on the net local feedback process. The mean east-to-west SST gradient amplifies the development of IOD through the zonal advection process. Anomalous easterly (westerly) wind during the positive (negative) IOD development carries anomalous warm water from east to west (west to east), and amplifies the positive (negative) IOD development. Thus, the decrease in the mean zonal SST gradient weakens this zonal advective feedback, and contributes to the weakening of the Bjerknes feedback processes.

However, as you pointed out, the term – ‘positive IOD-like change’ includes not only the change in the zonal SST gradient, but also the change in the thermocline depth and the wind pattern. Therefore, the ‘positive IOD-like change’ is related to various feedback processes, not only with the zonal advective process, but also with such as the thermocline-SST feedback. Such comprehensive terminology would cause confusion to the readers, especially when discussing the related feedback

processes. To clarify the meaning, we removed the term 'positive IOD-like change' in **Supplementary Discussion 3**.

We carefully reexamined **Supplementary Discussion 3** and found that our description of the Bjerkness feedback process coincides with the previous studies.

- In response to your concern, the shoaling thermocline, which is part of the 'positive IOD-like change', enhances the thermocline-SST feedback, and contributes to the strengthening Bjerkness feedback. This role of thermocline shoaling is described in **Supplementary Discussion 3** as "Shoaling of the thermocline depth increases the sensitivity of the thermocline depth to the zonal wind stress anomaly (Supplementary Fig. 29a). This change would intensify the thermocline-wind feedback". We revised the term 'thermocline-wind feedback' to 'wind-thermocline-SST feedback' to clarify the feedback loop chain in its term.

There is another description of the thermocline-SST feedback in **Supplementary Discussion 3**. The increased easterly wind, which is part of the 'positive IOD-like change' and associated thermocline depth gradient increase strengthens the thermocline-SST feedback and contributes to the increase in the Bjerkness feedback. It is described as "The increased climatological west-to-east thermocline depth gradient (Fig. 3b) due to the strengthened easterly wind would also enhance the thermocline-SST feedback as well."

- The 'positive IOD-like change' involves changes in multiple mean states including the zonal SST gradient, thermocline depth, and wind stress anomaly. These changes in mean states are related to the various parts of Bjerkness feedback processes. Some of them suppress the Bjerkness feedback, and some of them strengthen the Bjerkness feedback. For example, the decreased zonal SST gradient reduces the zonal advective feedback and contributes to the weakening of the Bjerkness feedback. In contrast, the decreased thermocline depth intensifies thermocline-wind-SST and thermocline-SST feedback processes and contributes to the strengthening of the Bjerkness feedback. The net changes in Bjerkness feedback are determined by the sum of these positive/negative effects. Therefore, the change

in each Bjerkness feedback does not necessarily follow the total net change in the IOD feedback process. Such offsetting effect was discussed in **Supplementary Discussion 3** and the previous studies.

5 The consensus of model results needs to be tested. The statistical significance needs to be ascertained.

e.g. L252 Furthermore, our study suggests the possibility of dramatic changes in the characteristics of IOD in a very warm climate environment, such as the occurrence of double peaks in the boreal spring and fall, and a zonally uniform oscillation pattern.

Are these results significant?

Reply: First, the L252 refers specifically to the CESM1.0.4 results, which are explained in the "**Equilibrium IOD response**" section of the paper. The aim of this line, which is placed in the discussion section, is to address the implication of this study rather than to explain the detailed result. In the high-CO₂ simulations of the CESM1.0.4, a seasonal double peak occurs in the boreal spring and fall, and the spatial pattern shifts from the dipole- to the monopole-like pattern. Although such specific transitions occur in only one model, it raises the possibility of a dramatic transition of IOD (i.e., critical transition). We believe that the dramatic transition in CESM1.0.4 is worth mentioning in the discussion section where an implication of this study is addressed. Given its motivation and purpose, we stated in the L252 "...suggest the possibility...".

To clarify that the results come specifically from the single model, we revised the L252 as "Furthermore, our study suggests the possibility of dramatic changes in the characteristics of IOD in a very warm climate environment, such as the occurrence of double peaks in the boreal spring and fall, and a zonally uniform oscillation pattern, as seen in the CESM1.0.4 simulations."

We assume that your concerns about the L252 are partly based on a

misunderstanding, as if it represents common results of the LongRunMIP models. However, as we explained, the L252 is only related to the single model (which is explained in the “**Equilibrium IOD response**” section). It is therefore not subject to the inter-model consensus test.

Following your comment, we performed a series of additional statistical significance tests on the LongRunMIP model results. This additional statistical significance test also covers the CESM1.0.4 results (i.e., the results in the L252) which you raised concerns about. We also performed additional analysis on the inter-model consensus ratio. The revised parts are as follows:

(1) Statistical significance

To measure the statistical significance of the equilibrium IOD amplitude change, we performed a two-sample F-test of equal variance. For each high-CO₂ simulation DMI, we performed the two-sample F-test with the control simulation DMI.

- IOD amplitude: In all high-CO₂ simulations, the equilibrium IOD amplitude change is statistically significant ($P < 0.05$). We accordingly added the result in the caption of **Fig. 1** as “In all high CO₂ simulations, the IOD amplitude change compared to the control simulation is statistically significant ($P < 0.05$). The statistical significance is tested with the F-test for a pair of DMI time series of the control and high CO₂ simulations.”

- IOD seasonal amplitude: We marked a seasonally significant point ($P < 0.05$) in **Supplementary Fig. 5**. The majority of the high-CO₂ simulation shows the significant seasonal IOD amplitude change in the peak season. We accordingly added the result in the caption of **Supplementary Fig. 5** as “The statistical significance of the change in the seasonal DMI standard deviation in the high-CO₂ simulation compared to the control simulation is tested with the F-test. The F-test is performed for a pair of DMI time series of the control and high-CO₂ simulations for each calendar month. The statistically significant ($P < 0.05$) point is marked as a

black circle.”

To measure the statistical significance of the CESM1.0.4 EOF pattern changes in the high-CO₂ simulations, we performed a two-sample F-test. The F-test is performed on a pair of the reconstructed second mode surface temperature anomalies of the control and high-CO₂ simulations for each grid point.

- EOF patterns of CESM1.0.4 simulations: We marked a statistically significant point ($P < 0.05$) in **Supplementary Fig. 7**. In abrupt2x, abrupt4, and abrupt8x simulations, the change is significant in the vast majority of the grid points. We accordingly added the result in the caption of **Supplementary Fig. 7** as “The statistical significance of the change in the second EOF mode surface temperature anomaly in the high-CO₂ simulation compared to the control simulation is tested with the F-test. The F-test is performed for a pair of the reconstructed second mode surface temperature anomalies of the control and high-CO₂ simulations for each grid point. The statistically significant ($P < 0.05$) point is marked as a black circle.”

The significance test on the seasonal IOD amplitude and CESM1.0.4 EOF patterns together resolve the concern you raised on the L252.

Additionally, to test the statistical significance of the long-term trend in the transient IOD amplitude, we calculated the p-value for the linear trend. We calculated the p-value for the transient long-term sensitivity of IOD amplitude, positive IOD event intensity, and negative IOD event intensity. The results are as follows:

- IOD amplitude: In all high-CO₂ simulations, the linear trends are statistically significant ($P < 0.05$). We accordingly added the result in the caption of **Fig. 2** as “All linear trends are statistically significant ($P < 0.05$).”

- Positive IOD event intensity: In all high-CO₂ simulations, the linear trends are statistically significant ($P < 0.05$) except for GISS-E2-R abrupt4x. GISS-E2-R abrupt4x shows a negative linear trend, but is not statistically significant ($P = 0.06$). When counting the number of high-CO₂ simulations with a decreasing trend, GISS-E2-R abrupt4x is excluded. We accordingly added the result in the caption of **Supplementary Fig. 9** as "All linear trends are statistically significant ($P < 0.05$) except for GISS-E2-R abrupt4x. GISS-E2-R abrupt4x shows a negative linear trend, but is not statistically significant ($P = 0.06$). When counting the number of high-CO₂ simulations with a decreasing trend, GISS-E2-R abrupt4x is excluded."
- Negative IOD event intensity: In all high-CO₂ simulations, the linear trends are statistically significant ($P < 0.05$) except for CCSM3 abrupt8x. CCSM3 abrupt8x shows a positive linear trend, but is not statistically significant ($P = 0.06$). We accordingly added the result in the caption of **Supplementary Fig. 10** as "All linear trends are statistically significant ($P < 0.05$) except for CCSM3 abrupt8x. CCSM3 abrupt8x shows a positive linear trend, but is not statistically significant ($P = 0.06$)."

(2) Inter-model consensus

In the main text of the original paper, we explicitly provided the inter-model consensus ratio for the equilibrium response for IOD amplitude (**L127**), positive IOD event intensity (**L129**), negative IOD event intensity (**L130**), IOD amplitude sensitivity (**L137**). We found that inter-model consensus ratios for the transient response are not explicitly provided in the original paper (although they can be found from the corresponding figure). We accordingly added them.

- We added the inter-model consensus ratio for the long-term sensitivity of IOD amplitude in **L175**. The added result is read as "In nearly all high-CO₂ simulations (17 out of 18), the IOD amplitude shows a long-term decreasing trend with increasing GMST. The exception is HadCM3L abrupt8x."
- We additionally performed analysis on the transient IOD event intensity changes

followed by Reviewer #2's comment (**Supplementary Figs. 9 and 10**). We accordingly added the inter-model consensus ratio for the long-term sensitivity of positive/negative IOD event intensity in **L202**. The added result is read as: "Consistent with the IOD amplitude change, the intensity of positive and negative IOD events also shows a long-term decreasing trend with increasing GMST (15 of 18 high-CO₂ simulations for positive IOD and 15 of 18 for negative IOD), but shows large fluctuations and widespread transient sensitivity to GMST (Supplementary Figs. 9 and 10)."

6 The exact meaning of some items, e.g. IOD amplitude, IOD variability, internal variability, transient IOD response, need to be further clarified, make them easier for readers to understand.

Reply: Following your suggestion, we clarified the following terms:

(1) IOD variability

We added a definition of IOD variability, where the term 'IOD variability' first appears in the manuscript in **L54**. The full sentence with the added definition reads as follows:

"IOD variability, a fluctuation in climate variables induced by the IOD, is typically measured by the dipole mode index (DMI), which is defined as the difference between the western (50° E-70° E and 10° S-10° N) and eastern (90° E-110° E and 10° S-0° N) SST anomalies of the tropical Indian Ocean."

(2) IOD amplitude and IOD event intensity

We added a clear explanation of 'IOD amplitude' and 'IOD event intensity' in **L105**. The added explanation reads as follows:

"We apply two different measures for the strength of the IOD variability. The main metric is the IOD amplitude, defined as the standard deviation of the DMI. The

other is IOD event intensity, defined as the peak DMI during the positive or negative IOD event (Methods). Both metrics are highly correlated by definition (i.e., if the IOD amplitude is high, the IOD event intensity is very likely to be high, and vice versa), but we mainly use the IOD amplitude to examine the change in the IOD variability strength. The IOD amplitude is a fundamental metric that can show the overall response of the IOD variability to greenhouse gas warming. The simplicity of the IOD amplitude definition has advantages that facilitate the physical interpretation of the results.”

(3) Equilibrium and Transient IOD response

We added a new section in Methods, which explains the exact definition of equilibrium and transient IOD response. As we briefly explain the equilibrium and transient response in the main text, we decided to place their exact definitions in Methods, considering the flow of the main text. Please see a newly added section entitled, ‘**Definition of the equilibrium and transient response**’ in **L409**. The section is read as follows:

“The equilibrium IOD response is defined as the response of IOD variability to an increased equilibrium GMST level. The equilibrium response reflects the forced equilibrium response due to greenhouse gas warming. The transient response is defined as the evolution of IOD variability in response to increasing GMST with time. The transient response reflects both the forced response and internal variability. Specifically, we interpret the long-term trend in IOD amplitude (or IOD event intensity) as the forced response, and the fluctuation in IOD amplitude (or IOD event intensity) that deviates from the long-term forced trend as the internal variability.”

(4) Internal variability

We added the exact meaning of ‘internal variability’ in the newly added section,

entitled '**Definition of the equilibrium and transient response**' in **L409**. The explanation in this part read as follows:

"Specifically, we interpret the long-term trend in IOD amplitude (or IOD event intensity) as the forced response, and the fluctuation in IOD amplitude (or IOD event intensity) that deviates from the long-term forced trend as the internal variability."

We also added the exact definition of the internal variability in the main text in **L184**. The added explanation read as follows:

"Throughout the paper, we refer to the anomalous short-term fluctuation in the variable that deviates from the long-term forced change as internal variability."

In addition to this newly added exact definition of IOD internal variability, we refer to the general meaning of the internal variability in the main text, such as in **L73** (general meaning of internal variability). We believe that this newly added explanation can make it easier to understand.

Author's Reply to the Reviewer #2

Reviewer #2 (Remarks to the Author):

Review for submission "Decreased Indian Ocean Dipole variability under prolonged greenhouse warming"

This study utilized LongrunMIP to address the forced long-term changes in the Indian Ocean Dipole (IOD) amplitude arising from natural variability. The manuscript is well-written, and the arguments are supported. Nevertheless, there are some concerns that the authors should address in the revision.

Reply: We appreciate your recognition and valuable comments on our paper. Thanks to your constructive comments, we have made a substantial improvement to the manuscript. The revised manuscript contains some major updates as follows:

1. We evaluated the performance of the LongRunMIP model and selected models that can reasonably simulate the IOD. We updated the analyses in the paper with these selected models (9 out of 11 models). For the model selection process and criteria, please see the newly added section **Supplementary Discussion 1**.
2. We performed the physical mechanism analysis on how ENSO influences IOD. Please see the newly added section **Remote ENSO influence on the IOD change**.

The reply to your comment is attached below. Please note that we will refer to the text line and figure/table number based on the revised version of the manuscript.

1. About the natural variability. Line 80: "IOD amplitude in the twenty-first century either increases or decreases depending on the pathways of internal variability, despite the same warming forcing." The term 'pathways' may be more closely associated with forcing rather than natural variability. Please consider revising for

clarity."

Reply: We agree that the term 'pathways' is not clear for this sentence. Following your suggestion, we remove the term 'pathways' and clarified the sentence as follows:

"A large ensemble simulation using the Community Earth System Model shows that internal variability alone can generate widespread long-term trends in future IOD variability change; projected IOD amplitude in the twenty-first century either increases or decreases depending on the ensemble member with different internal variability, despite the same warming forcing."

Thank you.

2. About the simple IOD model. In paper 20, four terms are discussed: local feedback, delayed feedback, ENSO impact, and stochastic forcing. In this study, the authors only consider three terms, excluding the delayed feedback. Please provide more discussions or explanations on this consideration. Line 381: "We consider the seasonality of λ , β , and σ . Thus, these three parameters vary seasonally. The typical value of the annual mean λ , estimated from observations and climate models in previous studies, is negative.' It is assumed here that the model is fitted for each month with ENSO and DMI index in phase (lag-0). Whether you consider a lead-lag relationship? Also, for the annual mean model, does the equation still hold?"

Reply: We would like to explain about the simple IOD model as follows:

(1) Delayed feedback term

As you pointed out, we did not explicitly consider the delayed feedback term. There are several versions of the simple IOD model. For example, Malte et al. (2017) and our study consider three terms, local feedback (λT), remote ENSO impact ($\beta \times \text{ENSO}$), and stochastic forcing ($\sigma \xi$). An et al. (2022) (which is written by our research group) additionally considers the delayed feedback term ($\alpha T(t-\tau)$) as you mentioned. Both

versions of the model are known to reproduce the DMI time series very well, and the difference comes from the consideration of the delayed feedback process.

To explain the difference between the two versions of the model, we would like to explain the physical meaning of $\alpha T(t-\tau)$. As the wind forcing is exerted in the Indian Ocean, the response of the thermocline to the wind forcing is achieved by the ocean waves. There are two waves involved in this process, the Kelvin wave and the Rossby wave. The Kelvin wave is fast; therefore, the Kelvin wave response of the thermocline and the associated SST are almost simultaneous, and this process can be included in λT . In contrast, the Rossby wave is three times slower than the Kelvin wave (time delay is typically 2 months), thus it is a delayed process. To explicitly account for this delayed Rossby wave response, An et al. (2022) introduced a new term, $\alpha T(t-\tau)$.

As we explained, Malte et al. (2017) and our study did not include this term in the simple IOD model, and only used λT to represent the feedback process occurring in the tropical Indian Ocean. The reason for this is that λT can implicitly account for the delayed Rossby wave response.

If we fit the model parameter for the simple IOD model with $\alpha T(t-\tau)$, the effect of the delayed Rossby wave response would be explicitly represented by $\alpha T(t-\tau)$. In contrast, if we fit the model parameter for the simple IOD model without $\alpha T(t-\tau)$, the effect of delayed Rossby wave response cannot be explicitly represented by $\alpha T(t-\tau)$. Instead, it would be implicitly included in the fitted seasonally varying λ . As an idealized example, we assume that the Rossby wave initiation activity peaks in October and the lag time is 2 months. For the model with $\alpha T(t-\tau)$, this process would be explicitly represented as $\alpha T(t-2)$ with α peaking in October. In contrast, for the model without $\alpha T(t-\tau)$, this process would be included in λ . Specifically, it would increase the value of λ in December. Therefore, λT can implicitly account for the Rossby wave response effect (at least partially).

The very high simulation skill of the simple IOD model demonstrated in our study and in Malte et al. (2017), shows that the exclusion of $\alpha T(t-\tau)$ is a reasonable

simplification that doesn't hinder the actual model performance. Moreover, the simplicity of the model without $\alpha T(t-\tau)$ also brings advantages that facilitate the physical understanding of the model results, especially in determining whether the IOD amplitude change is caused by a local process or a remote process.

To clarify this, we added the explanation in the main text. Please see **L474**. The added explanation is read as follows:

"We note that there are several versions of the simple IOD model in the current literature depending on the types of feedback processes included. The simple IOD model used in this study is the basic type model introduced in ref. ³¹, which considers the essential physical process of IOD in the simplest way. Hence, the model does not explicitly account for the time-delayed Rossby wave process as done in ref. ²⁰, and implicitly considers it by λT , which comprehensively represents local feedback processes in the tropical Indian Ocean. The simplicity of the model used in this study has advantages that facilitate the physical understanding of the model results, especially in determining whether the IOD amplitude change is caused by a local process or a remote influence."

(2) Lag between ENSO and DMI

The simple IOD model can consider the lag between the Niño 3.4 index and DMI. As you can read from the equation, the model relates the Niño 3.4 index (ENSO) with the tendency of DMI (dT/dt), not with the DMI itself (T). Therefore, Niño 3.4 index and DMI are not in-phase and would show an out-of-phase relationship.

We mathematically demonstrate that the model can consider the lag between Niño 3.4 and DMI. For simplicity, we consider only the ENSO forcing in the simple IOD model, $dT/dt = \beta \times \text{ENSO}(t)$. We set the Niño 3.4 index to vary sinusoidally with amplitude A and frequency w , $\text{ENSO}(t) = A \cos(wt)$. The analytical solution of T is $T(t) = (\beta A/w) \sin(wt)$. As you can see, ENSO and T show a phase lag of $\pi/2$. This is a highly idealized example showing that the basic form of the model (which relates

the Niño 3.4 index with the tendency of the DMI) can consider a lag between the Niño 3.4 index and the DMI. For the full model with a real Niño 3.4 index, the lag would depend on the λ , β , and characteristics of the Niño 3.4 index time series.

The model parameter fitting is the process of finding the optimal set of parameters that can best explain the given time series under the constraints of the model equation. Since the model itself can naturally consider the lag relationship, it is not necessary to modify the equation and the associated parameter fitting process to consider the lag. It is okay to just fit the model parameters to the original equation without modification.

To clarify that the model can consider the lag, we added an explanation in **L470**. The added explanation is read as follows:

“The model can consider the lag between ENSO and T because the model equation relates the ENSO with the tendency of T, not with the T itself”.

(3) Annual mean model

First of all, we would like to clarify the meaning of the ‘annual mean’ in this sentence “The typical value of the annual mean λ ...”. The ‘annual mean’ here, means the annual mean value of the seasonally fitted parameter, not the annually fitted parameter. We would like to note that the two are different. The former is obtained by (1) fitting the parameter for each season (12 calendar month value for each parameter), and (2) taking the annual mean value of the parameter. The latter is obtained by fitting the parameters for the entire period without considering seasonality (single value for each parameter).

In this paper, we only used and discussed the seasonally-varying model. We also only cite the reference that uses the seasonally-varying model, including the sentence you mentioned (as far as we know, all literature using the simple IOD model uses the seasonally-varying model to consider the seasonality of the physical process). The annual model (i.e., model with seasonally-fixed parameters) is neither

used nor mentioned. Thus, the discussion of the annual mean model is out of the subject of this study, because we neither use it, nor explain about it, nor cite the related material.

We assume that your concern stems from the misunderstanding that we mentioned the annual model in this sentence, whereas our intention was to explain the annual mean value of the seasonally-fitted parameter. We found the sentence has a dual meaning and can cause confusion to readers. We would like to apologize for the unclarity and any confusion you may have experienced.

To explain the exact meaning of this sentence, background knowledge of the non-autonomous linear ordinary differential equation needs to be provided. λ characterizes the linear stability of T . For the case where λ is constant (i.e., seasonally-fixed annual mean model, which was not used in this study), if λ is negative (positive), T decays (grows). For the case where λ is seasonally-varying (i.e., the seasonal model used in this study), the annual mean value of the seasonally varying parameter, characterizes the stability of T . If the annual mean value of seasonally-varying λ is negative (positive), T decays (grows). The first case is straightforward, can be understood by a simple eigenvalue analysis, and does not require additional explanation. However, the second case (the model used and explained in this study) is not straightforward and requires heavy mathematical calculations to prove. The mathematical proof of this argument can be found in Kim and An (2021) (Journal of Climate) (<https://doi.org/10.1175/JCLI-D-20-0495.1>) (although it is for ENSO and the model is two-dimensional, it presents an essentially mathematically identical system).

After careful consideration, we decided to remove the sentence, "The typical value of the annual mean λ , which was estimated from the observations and climate models in the previous studies, is negative^{20,31,32}. Hence, without external ENSO and stochastic forcing, the simple IOD model decays because of a negative growth rate.". Because, it requires a heavy and complex explanation of the mathematical background to properly deliver the information to readers, and it is not such an

important discussion that affects understanding results in our paper.

Thank you.

3. About the seasonal changes. Paper 19 addressed that the early IOD is more frequently observed under warming conditions. In this study, Line 124 mentions, 'The seasonal variability of IOD decreases in almost all seasons (Supplementary Fig. 5).' Moreover, Figure S5 illustrates that HadCM3L and a few other models peak early in JJA. It would be beneficial to delve into a more detailed discussion of this change. Lines 135-146: You might consider separating the discussions on seasonal changes and spatial patterns into two paragraphs for clarity.

Reply: We appreciate your insightful suggestion. As you noticed, some of the models including HadCM3L and CCSM3II, also show a shift of the seasonal peak to an earlier season, which is consistent with the findings from Sun et al. (2022) (although the warming scenario is different). In the main text, we added a discussion of this seasonal shift in **L148**. The added discussion read as follows:

"This alteration in the seasonality also occurs in other models including HadCM3L and CCSM3II. HadCM3L simulations show a transition in seasonal peak from October (control and abrupt2x) to July (abrupt4x and abrupt8x). CCSM3II simulations show a gradual shift of seasonal peak from September (control) to August (abrupt700ppm) to June (abrupt1400ppm). Such a pronounced increase in IOD variability in boreal summer is consistent with the previous study¹⁹ shows an increased occurrence of early positive IOD events by global warming under the IPCC's high-emissions scenario (although the warming scenario is different)."

We also separated the paragraph. Thank you.

4. About the relationship between changes in IOD and ENSO. Paper 28 addressed long-term ENSO changes based on LongrunMIP. It would be valuable to explore

whether models agree on changes in both IOD and ENSO and if the dominating mechanisms are similar. Authors should engage in discussions on this aspect.

Reply: We performed a series of additional experiments and analyses on the role of ENSO on the IOD. First, we performed the additional simple IOD model experiment to separate the contribution of the ENSO amplitude change to the IOD amplitude change (previously we quantified the contribution of the combined effect of ENSO variability and sensitivity of IOD to ENSO). Please see the newly added experiment result in **L245**. Second, we performed an analysis of the physical mechanism of the remote ENSO forcing on the IOD amplitude change. Please see the newly added section, "**Remote ENSO influence on the IOD change**" in **L260**. These two analyses deepen the understanding of the relationship between ENSO and IOD.

The ENSO amplitude decreases in all of the high-CO₂ simulations, consistent with the paper 28 you mention (**Supplementary Fig. 15a** and **Fig. R1**). The additional experiment shows that the forced decrease in ENSO amplitude has a very small contribution to the forced decrease in IOD amplitude. Therefore, the forced changes in IOD and ENSO amplitude are independently arising from greenhouse gas warming. The forced change in IOD amplitude is attributed to the change in the tropical Indian Ocean, and the forced change in ENSO amplitude is attributed to the change in the tropical Pacific Ocean (paper 28). Both ocean basins show similar features, positive IOD-like change and El Niño-like change, respectively. The inter-simulation correlation between forced ENSO and IOD amplitude change is 0.56 (**Fig. R1**). The correlation shows that there is some model agreement on the changes, but does not imply causality between them.

Fig. R1. Relationship between change in ENSO amplitude and IOD amplitude in the equilibrium period. The change is a percentage change from the control simulation value (colored dots). Each colored dot represents the results for each high-CO₂ simulation. The linear regression line is shown as a black line. The Pearson correlation coefficient is 0.56.

We have carefully considered including this part in the discussion section, in addition to the newly added analyses (L245 and “**Remote ENSO influence on the IOD change**”), but we concluded that it is somewhat out of the scope of the paper which focuses on the IOD. We feel that the analysis of the ENSO itself and the tropical Pacific Ocean should be done consistently with the IOD in order to confidently argue it, although the paper 28 provides a very relevant reference. We believe that such a detailed analysis of ENSO and the tropical Indian Ocean is beyond the scope of this paper. Furthermore, the relationship between ENSO and IOD is already elucidated in the newly added analyses. Therefore, we decided not to include this part in this paper. Thank you for your suggestion.

5. About the definition of IOD amplitude. Line 102: 'We measure the IOD amplitude as the standard deviation of the DMI.' Typically, the community uses the peak

season (SON) DMI to gauge IOD changes. Here, monthly DMI is employed; the authors may wish to explain this choice.

Reply: As you pointed out, the IOD change can be examined with the peak season (SON) DMI. In this study, we use two different metrics to examine IOD change. One is "IOD amplitude" which is defined as the standard deviation of DMI. The other one is "IOD event intensity" which is defined as the highest (or lowest) DMI level during the IOD event. Here, the positive (negative) IOD event is defined as a three-month running mean of DMI exceeding (below) the one standard deviation of the time series for at least three consecutive months. Therefore, the "IOD event intensity" is close to a typical metric for the IOD change (a peak season DMI).

These two metrics are highly correlated by their definition (i.e., if the IOD event intensity is high, the IOD amplitude is likely to be high, and vice versa). In this study, we analyze both metrics, but mainly focus on the "IOD amplitude".

There are several reasons for this. Most of all, the IOD amplitude is a more fundamental metric than the IOD event intensity. The IOD amplitude measures the overall variability of a warm-cold cycle of DMI, considering the entire SST cycle of the tropical Indian Ocean. Meanwhile, the IOD event intensity measures peak DMI, emphasizing the maximum SST variability. Therefore, the IOD amplitude is a more fundamental metric than the IOD event intensity, which can show more overall response of tropical Indian Ocean SST anomaly.

This facilitates the physical interpretation of the IOD change. We utilize the simple IOD model to disentangle the physical source of the IOD change. Here, the main variable of this model is DMI (T) (including this model, theory and model of IOD is typically based on DMI). Therefore, IOD amplitude, which is a standard deviation of DMI, can be directly linked with the simple IOD model, given by $((\langle T^2 \rangle - \langle T \rangle^2)^{1/2})$ where $\langle \rangle$ denotes ensemble mean (even an analytical solution of DMI standard deviation can be directly retrieved under very idealized parameter settings). Meanwhile, the IOD event intensity cannot be directly expressed with T as done for the IOD amplitude due to the complexity of its definition. Also, using IOD amplitude

is easier to directly attribute which region (eastern or western tropical Indian Ocean) contributed to the IOD change by simply measuring the standard deviation of SST of these two regions.

In summary, these two metrics are highly correlated and both show the response of IOD to greenhouse gas warming. We analyzed both metrics, but more focus on the "IOD amplitude" throughout the paper by its usefulness for the physical interpretation.

To clarify this point we made two changes in the manuscript:

(1) We added the additional explanation of the IOD amplitude definition. Please see **L105**. The added explanation is read as follows:

"We apply two different measures for the strength of the IOD variability. The main metric is the IOD amplitude, defined as the standard deviation of the DMI. The other is IOD event intensity, defined as the peak DMI during the positive or negative IOD event (Methods). Both metrics are highly correlated by definition (i.e., if the IOD amplitude is high, the IOD event intensity is very likely to be high, and vice versa), but we mainly use the IOD amplitude to examine the change in the IOD variability strength. The IOD amplitude is a fundamental metric that can show the overall response of the IOD variability to greenhouse gas warming. The simplicity of the IOD amplitude definition has advantages that facilitate the physical interpretation of the results."

(2) We additionally performed analysis on the transient IOD event intensity changes (previously there was only analysis for the equilibrium IOD event intensity change which was shown in **Supplementary Fig. 3**). Please see newly added **Supplementary Figs. 9 and 10**. As you can see, the change in positive and negative IOD intensity is consistent with the change in IOD amplitude. The majority of high-CO₂ simulation shows a decreasing trend in IOD event intensity against GMST (16 out of 18 for the positive IOD event and 15 out of 18 for the negative IOD event)

and shows large fluctuation. We also accordingly added an explanation in the main text. Please see **L202**. The added explanation is read as follows:

"Consistent with the IOD amplitude change, the intensities of both positive and negative IOD events also show long-term decreasing trends with increasing GMST (15 of 18 high-CO₂ simulations for positive IOD and 15 of 18 for negative IOD), but show large fluctuations and widespread transient sensitivity to GMST (Supplementary Figs. 9 and 10)."

6. About transient response. Line 95: For the 1pct2x and 1pct4x runs, 'After the CO₂ increase, the forcing is stabilized for typically longer than 1,000 years.' While the first 200 years may be considered the transient period, the runs predominantly reflect the equilibrated response after the forcing stabilizes. Line 148: 'We analyze the transient response of the IOD amplitude to the CO₂ forcing. We calculate the 100-year moving IOD amplitude for the entire simulation period.' It is noted that the entire period (1000 years or more) covers both the transient and equilibrated periods. The authors may need to consider taking the first 100-200 years as the transient period when the forcing is changing. After the forcing stabilizes, it may not be considered part of the transient period.

Reply: We carefully considered your suggestion and would like to explain in great detail why we use the entire simulation period to examine the transient response.

As you can see in **Supplementary Fig. 2**, the GMST, which is the representative indicator of planetary warming level, increases over time due to the CO₂ forcing. Although the detailed warming path is different for the model and the experiment, there is a common GMST evolution pattern in the high-CO₂ simulations. It increases rapidly in the early period, and then gradually adjusts to the equilibrium level. In the final period of the simulation, GMST increases only slightly and reaches the (quasi-) equilibrium level.

For the equilibrium response analysis, we defined the last 500 years of the

simulation as the "equilibrium period". We examined the equilibrium response of IOD by performing an inter-experiment comparison for the IOD amplitude (and event intensity) in the equilibrium period.

In contrast, for the transient response analysis, we used the entire simulation period and did not specifically define the 'transient period' (we do not use the term 'transient period' throughout the paper). Here, transient response refers to the temporal evolution of IOD variability (the strength of which is quantified as IOD amplitude and IOD event intensity) in response to greenhouse gas warming. Therefore, by definition, the entire simulation period shows the transient response of IOD to greenhouse gas warming. The continuous evolution of IOD variability over the whole simulation period shows the transient response of IOD during the period where both large (e.g., 0-200 years) and small (e.g., 2000-3000 years) increases in GMST occur. We would like to note that 'the time evolution of the IOD variability during the equilibrium period' and 'the response of IOD variability to increasing equilibrium GMST' are different. The former is the transient response of IOD by definition, and the latter is the equilibrium response of IOD that we analyzed in the first part of the paper.

As you suggested, there is an alternative approach to understanding the transient response for LongRunMIP simulations. We can define the transient period as the first 100-200 years (or the first 0-200 years) as you suggested. This type of approach considers the IOD amplitude change when the GMST (or forcing) change is large.

The difference between these two approaches lies fundamentally in how we define 'transient' and related terms. The approach of your suggestion implicitly defines a 'transient period' as the period during which the significant change in GMST (or CO₂ forcing) occurs. Therefore, it focuses on the change in IOD variability during the period when a large increase in GMST occurs. In contrast, our approach considers the 'transient response' as the temporal evolution itself. Therefore, our approach covers the change in IOD variability for the entire period, including the period when GMST increases small.

As you can see in **Fig. 2**, we analyzed the transient IOD response on the basis of GMST level, not time. This GMST-based analysis can naturally consider the difference in GMST level depending on the time period (the concern you raised). Since the changes in IOD-related variables are analyzed on the basis of GMST change, the transient IOD response during the period of large GMST increase period (e.g., 0-200 years), will make a large contribution to the result, while the transient IOD response during the period of small GMST increase period (e.g., 2000-3000 years), will make a relatively small contribution. Therefore, we can expect that, there would be no significant difference in the main results even if we perform the analysis for the selected transient period of large GMST increase.

To examine this, we calculate the IOD amplitude for the first 0-500 years for all simulations. The period 0-500 years is roughly a median value of the period of large GMST increase (e.g., GFDLCM3 1pct2x shows large GMST increase roughly up to year 1000 and MIROC 3.2 1pct2x shows up to year 200) (please note that there is a large difference in the GMST increase rate between simulations). Please see **Fig. R2**. As you can see, there is practically no difference in the main result when we choose the first 0-500 years for the transient response. We reach the same conclusion that the decreasing forced long-term trend in IOD amplitude occurs in the vast majority of the simulations.

Fig. R2. Same as **Fig. 2a** in the main paper, but for the transient period (years 0-500).

As we demonstrated, both approaches to ‘transient response’ analysis are reasonable and bring no practical difference to the main conclusion of the study. However, our approach has several advantages. First of all, we can avoid the problem of period selection dependence. If we specifically choose the transient period, we have to specifically set the period criteria (e.g., whether we choose the 50-150 years or the 10-200 years for the transient period). The result would depend on how we choose the transient period (although the difference may be small). Also, the transient period approach can potentially hinder the fair comparison due to the large inter-model spread in the rate of GMST increase. For example, MIROC32 1pct2x shows fast adjustment to equilibrium GMST level (proper transient period would be 0-200 years), while the GFDLCM3 1pct2x shows very slow adjustment to equilibrium level (proper transient period would be 0-1000 years). Therefore, it is

not fair to compare them in the same line by applying uniform transient period criteria. We also considered setting the 'transient period' criteria based on the rate of GMST increase, but this would be more complex and would hinder the interpretation of the results (somehow, we have to set specific criteria for the warming rate as well). In addition, our approach provides a complete description of the IOD amplitude evolution spanning the entire simulation period (the evolution during the transient period is part of our result using the entire period).

Therefore, our approach, which considers the entire simulation period, is the most natural approach to analyzing the time evolution of IOD in response to greenhouse gas warming. It allows a fair comparison between simulations with different rates of GMST increase and facilitates the understanding of the results in as consistent a framework as possible.

In addition, we would like to note that it would make more sense to use the GMST as the basis for analysis rather than the atmospheric CO₂ concentration level. Physically, increasing atmospheric CO₂ alters radiative forcing, and leads to planetary warming and alteration of the hydrological cycle. Therefore, this warming and associated changes are key physical effects of the CO₂ forcing. As there is a large time lag between CO₂ and GMST due to the inertia effect, it would be more physically reasonable to use GMST, which brings practical impact to the climate system, as the basis of the analysis.

For these reasons, after serious and careful consideration, we decided to retain the original analysis using the entire simulation period. To clarify our approach, we added a new section in Methods, which explains the exact definition of transient IOD response. Please see a newly added section entitled, '**Definition of the equilibrium and transient response**' in **L409**. The section is read as follows:

"The equilibrium IOD response is defined as the response of IOD variability to an increased equilibrium GMST level. The equilibrium response reflects the forced equilibrium response due to greenhouse gas warming. The transient response is defined as the evolution of IOD variability in response to increasing GMST with

time. The transient response reflects both the forced response and internal variability. Specifically, we interpret the long-term trend in IOD amplitude (or IOD event intensity) as the forced response, and the fluctuation in IOD amplitude (or IOD event intensity) that deviates from the long-term forced trend as the internal variability.”

Thank you for the suggestion.

7. About the period of analysis. Line 107: 'last 500 years.' Are the entire 500 years used for detrending through quadratic fitting? Line 312: 'For the LongRunMIP, we applied quadratic detrending to the monthly SST to remove the forced long-term change (i.e., secular mode).' Could the authors clarify whether they used a 100-year or 500-year period for this detrending process?

Reply: We apologize for the inconvenience you might have experienced due to the unclarity of the description. The detrending is performed for the entire period of the simulation (see **Supplementary Table 1** for the length of each simulation). Therefore, we did not use either the 100-year or 500-year period for the detrending as you questioned. This detrending removes the long-term warming trend induced by CO₂ forcing (please see **Supplementary Fig. 2** as an example of the long-term warming trend).

As a result, for each simulation, we obtain the eastern/western SST anomaly and DMI time series, the length of which is equal to the length of the entire simulation. For the equilibrium response analysis, we select the last 500 years of the DMI time series and calculate its standard deviation (results shown in **Fig. 1a**). For the transient response analysis, we calculate the 100-year moving DMI standard deviation for the entire period of DMI time series (results shown in **Fig. 2a**).

To clarify this process, we added explanations in **L388** and **L394**. The added explanations are read as follows:

“The detrending is performed for the entire period of the simulation (see

Supplementary Table 1 for the length of each simulation).”

“As a result, for each simulation, we obtain the DMI and Niño 3.4 index time series, the length of which is equal to the length of the entire simulation. Using these DMI and Niño 3.4 index time series, we perform analyses including the calculation of IOD amplitude and event intensity.”

Thank you.

8. Figure S13. The upper panel x-y axis should be swapped to be consistent with the lower panel.

Reply: We appreciate your suggestion and carefully considered the swapping the x-y axis in the upper panel.

Usually for a scatter plot, if the variable 'A' potentially causes variable 'B', 'A' is plotted on the x-axis and 'B' is plotted on the y-axis. In our case, the upper panel shows the relationship between change in GMST and change in the model parameter (λ , β , ENSO amplitude, σ). If there is no potential causal relationship between these two variables, it would be reasonable to swap the x-y axis and make it consistent with the lower panel, following your suggestion. However, there is a very strong potential causal relationship between them. Physically, the change in model parameters would be caused by the change in GMST (i.e., forced warming by the CO₂ increase). Therefore, considering the strong causal relationship between them, it is more reasonable to plot the GMST change on the x-axis and the model parameter change on the y-axis.

The same plot style is applied to the lower panel. The lower panel shows the relationship between the change in the model parameter (λ , β , ENSO amplitude, σ) and the change in the DMI standard deviation. As we can read from the simple IOD model equation, the change in the model parameter directly causes changes in the DMI and its standard deviation. It is therefore reasonable to plot the GMST change on the x-axis and the model parameter change on the y-axis.

In conclusion, the current upper and lower panel shows a causal relationship between two variables in a consistent way (cause variable on the x-axis and effect variable on the y-axis). We believe maintaining the plotting style can deliver the results of this plot to readers emphasizing the physical relationship between them.

To clarify the causal relationship between the two variables, we added the following explanations in the figure legend:

“The GMST change by CO₂ forcing would cause the change in the model parameters and ENSO, contributing to the IOD amplitude change.”

Thank you.

REVIEWERS' COMMENTS

Reviewer #1 (Remarks to the Author):

The authors have responded to all my concerns and revised their manuscript accordingly. I have no further concerns or comments.

Reviewer #2 (Remarks to the Author):

The authors have effectively addressed my concerns by providing additional evidence and enhancing the clarity of the study. At this point, I have no further comments to add.